# The evolution of strategy in bacterial warfare via the regulation of bacteriocins and antibiotics

Rene Niehus[1], Nuno M Oliveira[2,3], Aming Li[4,5], Alexander G Fletcher[6,7], Kevin R Foster[8,9]*

[1]Center for Communicable Disease Dynamics, Harvard TH Chan School of Public Health, Harvard University, Boston, United States; [2]Department of Applied Mathematics and Theoretical Physics, University of Cambridge, Cambridge, United Kingdom; [3]Department of Veterinary Medicine, University of Cambridge, Cambridge, United Kingdom; [4]Center for Systems and Control, College of Engineering, Peking University, Beijing, China; [5]Institue for Artificial Intelligence, Peking University, Beijing, China; [6]School of Mathematics and Statistics, University of Sheffield, Sheffield, United Kingdom; [7]The Bateson Centre, University of Sheffield, Sheffield, United Kingdom; [8]Department of Zoology, University of Oxford, Oxford, United Kingdom; [9]Department of Biochemistry, University of Oxford, Oxford, United Kingdom

*For correspondence:
kevin.foster@zoo.ox.ac.uk

Competing interests: The authors declare that no competing interests exist.

**Abstract** Bacteria inhibit and kill one another with a diverse array of compounds, including bacteriocins and antibiotics. These attacks are highly regulated, but we lack a clear understanding of the evolutionary logic underlying this regulation. Here, we combine a detailed dynamic model of bacterial competition with evolutionary game theory to study the rules of bacterial warfare. We model a large range of possible combat strategies based upon the molecular biology of bacterial regulatory networks. Our model predicts that regulated strategies, which use quorum sensing or stress responses to regulate toxin production, will readily evolve as they outcompete constitutive toxin production. Amongst regulated strategies, we show that a particularly successful strategy is to upregulate toxin production in response to an incoming competitor's toxin, which can be achieved via stress responses that detect cell damage (competition sensing). Mirroring classical game theory, our work suggests a fundamental advantage to reciprocation. However, in contrast to classical results, we argue that reciprocation in bacteria serves not to promote peaceful outcomes but to enable efficient and effective attacks.

## Introduction

Bacteria commonly live in dense and diverse communities where competition for space and nutrients can be intense (*Kim et al., 2014*; *Hibbing et al., 2010*). As a response to such ecology, bacteria have evolved a wide range of competitive traits (*Granato et al., 2019*), including contact-dependent inhibition (*Ghequire and De Mot, 2014*; *Hayes et al., 2010*; *Aoki et al., 2010*), the type VI secretion system (*Miyata et al., 2013*; *Russell et al., 2014*; *Ho et al., 2014*; *Hood et al., 2010*), narrow-spectrum bacteriocins, and broad-spectrum antibiotics (*Bérdy, 2005*; *Riley and Wertz, 2002*; *Chao and Levin, 1981*), which can kill or inhibit other strains. These mechanisms are extremely widespread. Bacteriocidal toxins are found in almost all major bacterial lineages (*Riley and Wertz, 2002*; *Granato et al., 2019*) and single species commonly make use of multiple toxins and diverse means of attack (*Granato et al., 2019*; *Be'er et al., 2010*; *Zhang et al., 2012*; *Steele et al., 2017*;

*Jamet and Nassif, 2015*). This toxin-based warfare is also important for bacterial ecology and evolution, with evidence that toxin production can prevent competing strains from invading a niche (*Sassone-Corsi et al., 2016*; *Nakatsuji et al., 2017*; *Zipperer et al., 2016*; *O'Sullivan et al., 2019*), kill off coexisting strains (*Granato et al., 2019*; *Majeed et al., 2011*; *Speare et al., 2018*; *Ma et al., 2014*), or help strains to invade new niches (*Kommineni et al., 2015*; *Wiener, 1996*; *Sana et al., 2016*).

The production and regulation of bacterial toxins have been studied for decades because of their potential as clinical antibiotics (*Lewis, 2013*; *Slattery et al., 2001*). This work has revealed that toxin production is often tightly regulated (*Miyata et al., 2013*; *Anderson et al., 2012*; *Ghazaryan et al., 2014*; *Bernard et al., 2010*). Indeed, it is thought that there are many new antibiotics that remain undetected because they are only activated under certain conditions (*Maldonado et al., 2003*; *Abrudan et al., 2015*; *Traxler et al., 2013*). A major form of regulation in bacteria is quorum sensing (*Fuqua et al., 1994*; *Navarro et al., 2008*; *Eickhoff and Bassler, 2018*) whereby cells secrete a small molecule and respond to it dependent upon its concentration. Some antibiotics and bacteriocins are regulated by quorum sensing, which is thought to ensure that toxin production occurs at the right cell density (*Hibbing et al., 2010*; *Chandler et al., 2012*). Other factors also regulate bacterial toxin production, including particular nutrient conditions and diverse stress responses (*Storz and Hengge, 2011*). This led to the argument that, in addition to quorum sensing, bacteria engage in 'competition sensing' whereby they use nutrient stress and cell damage to detect ecological competition (*Cornforth and Foster, 2013*; *Lories et al., 2020*).

Bacteria, therefore, have the potential for a wide range of responses during combat. Evolutionary theory has so far focused on the evolution of unregulated toxin production. This work has highlighted that factors such as strain frequency, nutrient level, the level of strain mixing (relatedness), and the cost of toxin production are all important for whether bacteria employ toxins at all (*Bucci et al., 2011*; *Gardner et al., 2004*; *Brown et al., 2006*; *Gordon and Riley, 1999*; *Frank, 1994*; *Levin, 1988*). Other models have highlighted how natural selection for warfare can have consequences for the evolution of diversity (*Frank, 1994*; *Biernaskie et al., 2013*; *Kelsic et al., 2015*; *McNally et al., 2017*), including via rock-paper-scissor dynamics between different genotypes (*Czárán et al., 2002*; *Kerr et al., 2002*). However, to understand the strategic potential of warring bacteria, we must consider the regulation of their toxins and other weapons (*Granato et al., 2019*; *Cornforth and Foster, 2013*).

Here, we study the evolution of strategy during bacterial warfare by combining a detailed differential equation model of toxin-based competition with game theory to identify the most evolutionarily successful strategies. Informed by the large empirical literature on factors that regulate bacteriocins and antibiotics, we compare four major classes of potential strategies: constitutive (unregulated) toxin production, and regulation via nutrient level, quorum sensing, or by damage from a competitor's toxin. We study the behaviours and competitive success of each strategy when in competition with other strains across a range of scenarios. We find that all three types of regulated strategies carry benefits relative to non-regulated production and, for short-lived resources, the three types of regulation offer largely equivalent alternatives for controlling attacks. However, for long-lived environments, responding to incoming attacks is often the best performing strategy. A key benefit to such reciprocation in such environments is the ability to downregulate a toxin once a competitor is defeated, thereby saving the energy that would be lost in needless aggression.

## Results and discussion

### Overview

We are interested in how competition between strains and species of bacteria shapes the evolution of toxin regulation. The core of our approach is a set of detailed ordinary differential equations (ODEs) that capture ecological competition between bacteria (*Figure 1a*), which are built upon an earlier model of bacterial siderophore production (*Niehus et al., 2017*). After exploring the case of constitutive toxin production only, we extend our model to incorporate different strategies of regulated production (*Figure 1b*). We use these differential equations to model ecological interactions of bacterial strains and determine the outcome of competition for a given strategy against another strategy when they meet locally.

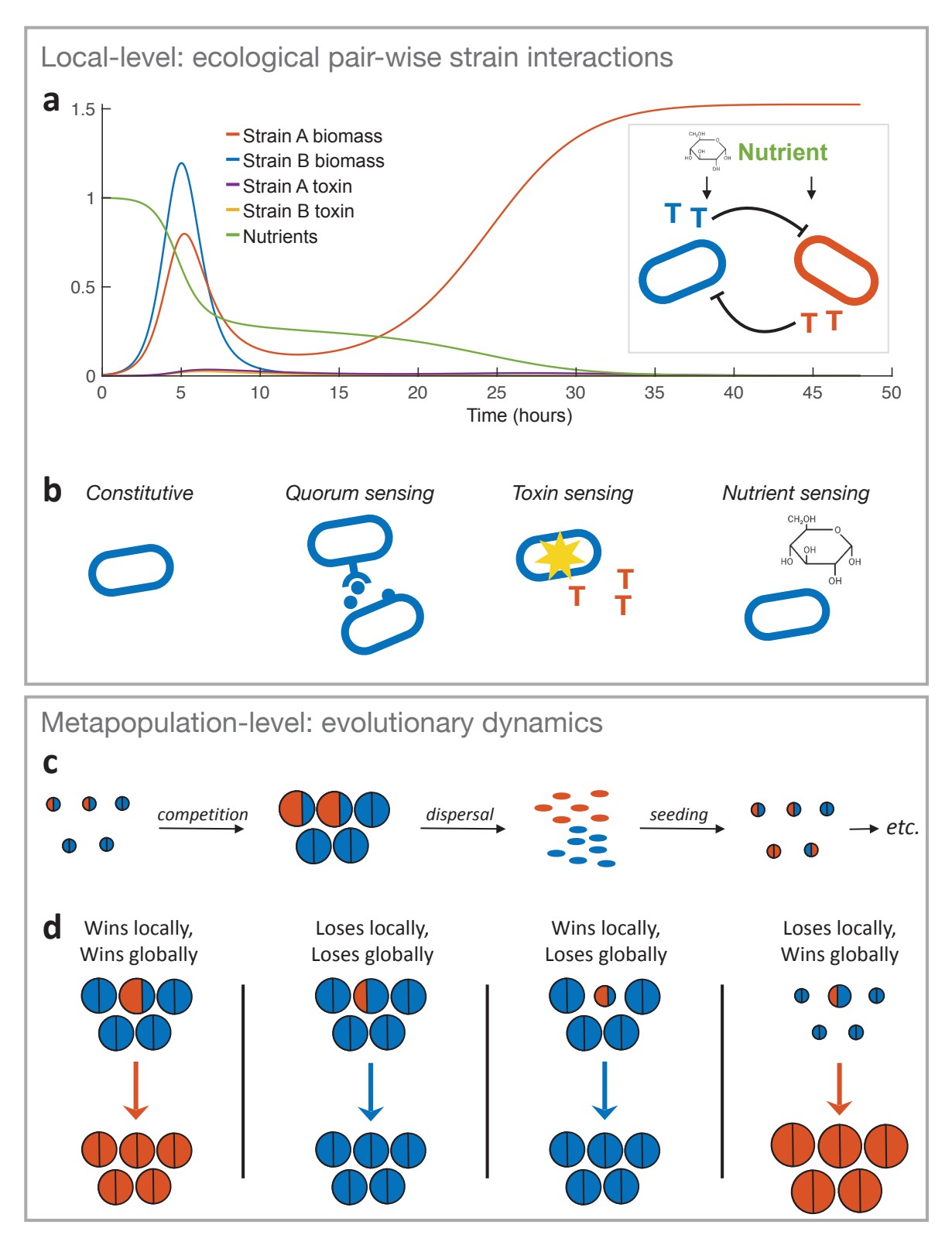

**Figure 1.** The two-layer modelling framework. (a) At the ecological timescale, we use differential equations to model the pairwise interactions of strains with competing strains represented here by two single cells in blue and orange. Both strains consume nutrients from a shared pool, and each strain can produce a toxin that inhibits the other strain (represented as coloured 'T's). We show an example of the temporal dynamics of a competition between two strains, where strain A wins by investing more into toxin production ($f_A = 0.3$) than strain B ($f_B = 0.1$). All other parameters take the standard values
*Figure 1 continued on next page*

*Figure 1 continued*

given in *Table 1*. (b) The differential equation model is used to model four major classes of toxin production strategies. From left to right: Constitutive production without sensing of the environment, sensing clone-mate density (quorum sensing), sensing damage by the competitor's toxin, and nutrient sensing. *Lower panel*: At the metapopulation level, we model the long-term evolutionary dynamics of different warfare strategies. (c) Bacterial life cycle assumed for modelling: empty patches are seeded with a small number of cells that then compete, where the outcome is determined according to the local-level model (above). Cells of the two different strategies are shown in blue and orange as circles, where the area represents the number of cells each produces. After competing in the patch for a certain amount of time (24 hours by default), the cells disperse, where the number of cells produced by each strategy determines its frequency in the dispersal phase and new patches. That is, all dispersing cells have the same probability of finding and seeding a new patch, and environmental conditions are identical across patches. Then another competition phase begins and so on (here orange is winning and invading the population). While we show only two different strategies here, we model a metapopulation with more than two strategies when we study the coevolution of attack strategies. (d) Four key outcomes used to predict evolutionary invasion. First case: a rare mutant outcompetes the resident strategy in its patch (orange area is bigger than blue area in the patch). Importantly, the mutant also wins globally, that is, it makes more cells than the *average resident* in the population, which we take from the number of cells that the resident strategy makes when it meets the *same* strategy (the size of a semicircle in the all-blue patches). This measure captures resident fitness well because with the mutant being rare and a large number of patches, the resident will nearly always be meeting itself. Second case: mutant loses and, in doing so, makes fewer cells than the average resident. Third case: mutant wins locally, but ends up making very few cells, for example, it redirects a lot of energy into toxins rather than growth. As a result, it does not produce more cells than the average resident strain (i.e. orange area in focal patch is smaller than blue area in all-blue patches). Fourth case: mutant loses locally but produces more cells than the average resident, for example, the mutant is more passive and avoids the strong mutual inhibition of two toxin producers. Thus the mutant wins globally.

These local-level competitions are embedded into a larger metapopulation framework that determines long-term evolutionary outcomes (*Maynard Smith, 1982*; *Figure 1c and d*). This metapopulation modelling includes invasion analysis, in the tradition of the branch of evolutionary game theory developed by *Maynard Smith and Price, 1973* and the field of adaptive dynamics (Materials and methods). We also later use a more explicit genetic algorithm that employs the same logic. This algorithm pits diverse strategies against each other across a large number of combinations in order to find the most successful strategies (Materials and methods). In these metapopulation models, bacterial strains are assumed to compete locally in a large number of patches, but also globally through dispersal to seed new, empty patches based on a standard life history of bacteria used in previous models (*Oliveira et al., 2014*; *Gardner et al., 2007*; *Chuang et al., 2009*; *Cremer et al., 2012*; *Nadell et al., 2010*; *Figure 1c*). Also, as discussed previously (*Nadell et al., 2010*), we refer to the global population as a metapopulation to distinguish it from the local bacterial cell population in each patch. This approach accounts for the possibility that a strategy can do well in local competition, but do poorly globally, and vice versa (*Figure 1d*). We show in later analysis that all competitive outcomes shown in *Figure 1d* occur in our simulations, with the first two cases being the most common.

We explore a number of different evolutionary scenarios using different calculations. We start by using invasion analysis (depicted in *Figure 1d*) to study the evolution of toxin producers that lack regulation and to study the evolution of toxin regulation from constitutive producers. This allows us to understand first when, and how much, a strain should invest into attacking other, and then, whether regulated production can evolutionarily replace constitutive production. We next compare different regulated strategies to one another by studying their performance when facing a diverse range of constitutively producing species. Finally, we study the case where regulated strategies compete with each other and coevolve in massive tournaments to identify the most globally successful strategies (see Materials and methods).

Our model needs to be relatively complex in order to capture the evolution of bacterial competition and regulatory networks. As a result, the form of our mathematical model is of a class that is not amenable to analytical work (*Boys et al., 2008*; *Gutiérrez and Rosales, 1998*; *Liu and Chen, 2003*). To confirm this, we investigated the behaviours of the dynamic model at steady state. This showed a good basic correspondence between our numerics and analytics but confirmed that the model is not amenable to further analytical work (see Appendix 1 Supplementary analytics). Nevertheless, by combining a number of different competition scenarios with wide parameter sweeps, we are able to show that our key conclusions are robust across many conditions.

## Evolution of warfare via unregulated toxin production

We first ask, what favours the evolution of constitutive toxin production. While many toxins are regulated, constitutive production does occur (*Mavridou et al., 2018*), and we use the simple case of constitutive toxin production to first identify general principles underlying the evolution of bacterial warfare. In addition, constitutive production forms a baseline from which to compare the evolution of regulated strategies. In order to study the behaviours that result from each strategy, we use a detailed model of competition between strains based upon a system of differential equations (Materials and methods). This approach allows us to capture the temporal dynamics of strain interactions and, later, toxin regulation.

In the model, we follow nutrient concentration and cell biomass over time as the strains engage with each other (*Figure 1a*). We focus on competitions between two strains that each possess a toxin that does not harm the producer strain but does harm the other strain. In reality, strains may carry multiple toxins and resistances (*Cordero et al., 2012*; *Gordon et al., 1998*) and our framework can be extended to include such complexity. However, for simplicity, we focus here on a single toxin produced by each strain. We consider interactions that are pairwise at the strain level, but we later account for a multitude of competitors by letting strains have many encounters, each with a different strategy. To enable us to study a large number of strategies, our differential equations are based upon simplified well-mixed conditions.

Our goal is to understand the evolutionary fate of different strategies of toxin-mediated competition. In order to do this, we need to recognise that the outcome of competition at a local scale may not be predictive of evolutionary trajectories. Consider, for example, a competition between two strains of bacteria on a particle of detritus in a pond. If one focuses solely on local competition on the particle, then any strategy that results in a focal strain making more cells than its competitor will be favoured, even if this leads to relative ruin for the winning strain with only a few cells surviving the process. However, given these competitions can happen on many such particles, it is unlikely that such extreme strategies would be favoured, because few cells will be produced to colonise new particles. Instead, the best strategies will be those that *make the most cells to disperse*, which may mean a strain also wins locally, but it may not (see *Figure 1d*).

To capture this effect, we embed the local-level competitions within a broader framework in order to make evolutionary predictions (*Maynard Smith, 1982*; *Weibull, 1997*; *Mitri et al., 2011*) (see *Figure 1c and d*). This framework allows us to ask whether a particular, initially rare, strategy can successfully invade a metapopulation of another strategy (Materials and methods). Specifically, a rare mutant's fitness in the metapopulation is defined by the number of cells it produces in direct competition with the resident, while the resident's fitness is defined by its productivity when it meets another resident in a patch, as will occur in the vast majority of patches if the mutant is rare (*Figure 1d*, Materials and methods). For mutants that can invade, we also confirm that they then cannot be reinvaded by the previous resident (Materials and methods), which is indeed always the case here. We refer to such invasions that lead to a full replacement of the resident by the mutant – where the resident is unable to reinvade from rare - as a stable invasion. By studying large numbers of competitions, we can categorise strains by their ability to stably invade others, and thereby identify the evolutionarily stable investment into toxin production (*f\**). We then seek the optimal level of toxin production, which cannot be invaded by any mutant strategy, but can invade all others.

What determines the optimal level of toxin investment? Intuitively, we find that cells evolve to invest more in attacking their competitors when toxins are efficient at killing the competitor and/or

**Table 1.** Model parameters and their effect on optimal toxin investment.

| Model parameter | Parameter description | Standard value [unit] (notes) | Effect on optimal toxin investment $f\*$ |
|---|---|---|---|
| $C(t=0)$ | Initial cell biomass of each strain | 0.1 [gC] | ↑ |
| $N(t=0)$ | Initial pool of nutrients | 1 [gN] | Intermediate optimum (*Appendix 1—figure 3*) |
| $K_N$ | Saturation constant for nutrient uptake | 5 [gN] | ↑ |
| $\mu_{max}$ | Maximum growth rate | 10 [1/hr] | ↓ |
| $k$ | Killing efficiency of the toxin | 20 [1/gT\*1/hr] | ↑ |
| $l_T$ | Toxin loss rate | 0.1 [1/hr] | ↓ |

the toxins persist stably in the environment (*Table 1*). Toxin efficiency in our model is equivalent to the relative cost of toxin production, that is, we see a high benefit-to-cost ratio favours toxin use. This result is in line with previous theory, which has shown that the impact of toxin production on growth rate is critical for the evolutionary outcome (*Bucci et al., 2011*; *Levin, 1988*). For highly effective toxins, we find that strains will engage in an arms race that escalates to the point where populations can go extinct (*Appendix 1—figure 4*). Such 'evolutionary suicide' is known from a wide range of conflict scenarios in biology (*Rankin and López-Sepulcre, 2005*).

While earlier models have studied the effects of nutrients on toxin production, these studies either did not model nutrients explicitly (*Frank, 1994*), or the level of nutrient competition was coupled to the presence of and mixing with other strains (*Bucci et al., 2011*; *Gardner et al., 2004*). In our model we can isolate the effect of nutrients on the evolution of toxin use. When nutrients are scarce, there is not enough energy to produce effective amounts of toxins (*Appendix 1—figure 3*), which agrees with previous theory (*Bucci et al., 2011*; *Gardner et al., 2004*; *Frank, 1994*), and has also been shown experimentally in yeast (*Wloch-Salamon et al., 2008*). However, we also find that toxin benefit peaks at intermediate nutrient availability and decreases for higher nutrient levels (*Appendix 1—figure 3*). This can be understood in terms of a shift in the relative benefits of investing in cell division versus attack: When bacteria enter a competition at low density and resources are abundant, there is a great potential for population expansion. Under these conditions, cells evolve to invest relatively little in toxin production; energy is instead better invested in rapid growth to win a competition by outgrowing other strains. In contrast, when growth potential is limited, cells benefit from investing in warfare, unless, as mentioned above, nutrients are too scarce to produce an effective toxin concentration.

## The evolution of regulated attack strategies

We next investigate what happens when cells are able to regulate their level of toxin production in response to environmental cues. The production of antibiotics and bacteriocins is commonly tightly regulated by a variety of signals and cues. As discussed above, these can be broadly divided into three major classes based upon known bacterial regulatory networks. The first is detection of cell density by canonical quorum sensing or related means (*Fuqua et al., 1994*; *Eickhoff and Bassler, 2018*), which has been demonstrated by previous modelling work to be beneficial for the regulation of cooperative traits (*Cornforth et al., 2012*). In addition, bacteria are highly responsive to both nutrient stress and cell damage associated stress (*Storz and Hengge, 2011*), which both can detect the level of ecological competition in the environment ('competition sensing'; *Cornforth and Foster, 2013*; *Lories et al., 2020*).

We first compare the evolution of regulation by quorum, nutrient level, and the level of the competitor's toxin when each is in competition with constitutive strains. This allows us to ask whether regulated strategies can evolutionarily replace constitutive strategies (see Materials and methods). In brief, we model regulation of toxin production using a simple step function, which is defined by toxin production in activated state ($f_{induced}$), production in inactivated state ($f_{initial}$), and a threshold of the signal for activation. All three parameters are continuous; toxin production ($f_{initial}$ and $f_{induced}$) is constrained between 0 and 1, and the threshold is constrained to a region consistent with the observed range of each signal (quorum, nutrient, toxin level).

In a vast tournament consisting of millions of individual competitions, we pit all possible strategies of each mode of regulation against all possible versions of the fixed strategy. We then use invasion analysis, as before, to look for the evolution of regulated strategies that can invade all unregulated strategies. As before, we consider both global and local competition (see *Figure 1d*) to determine invading strategies that cannot be reinvaded by the previous resident strategy and that therefore cause stable invasion. We find that all possible outcomes of the metapopulation competitions (*Figure 1d*) do occur, with the typical case being that the outcome of local and global competition are the same (see *Appendix 1—figure 6*). In a small minority of cases (2.3%), we find that successful invading strategies can be reinvaded by the previous resident to give a mixed evolutionary outcome, and these cases are not considered further. Our analysis identifies versions of each mode of regulation that can stably invade all possible constitutive strategies (*Figure 2a–c*). This result is expected and confirms the basic intuition that – unless maintaining a regulatory circuit is very costly – a well-regulated trait will outcompete an unregulated one (*Cornforth and Foster, 2013*). This is true whether strains compete for a short or long duration, although shorter duration does select for a

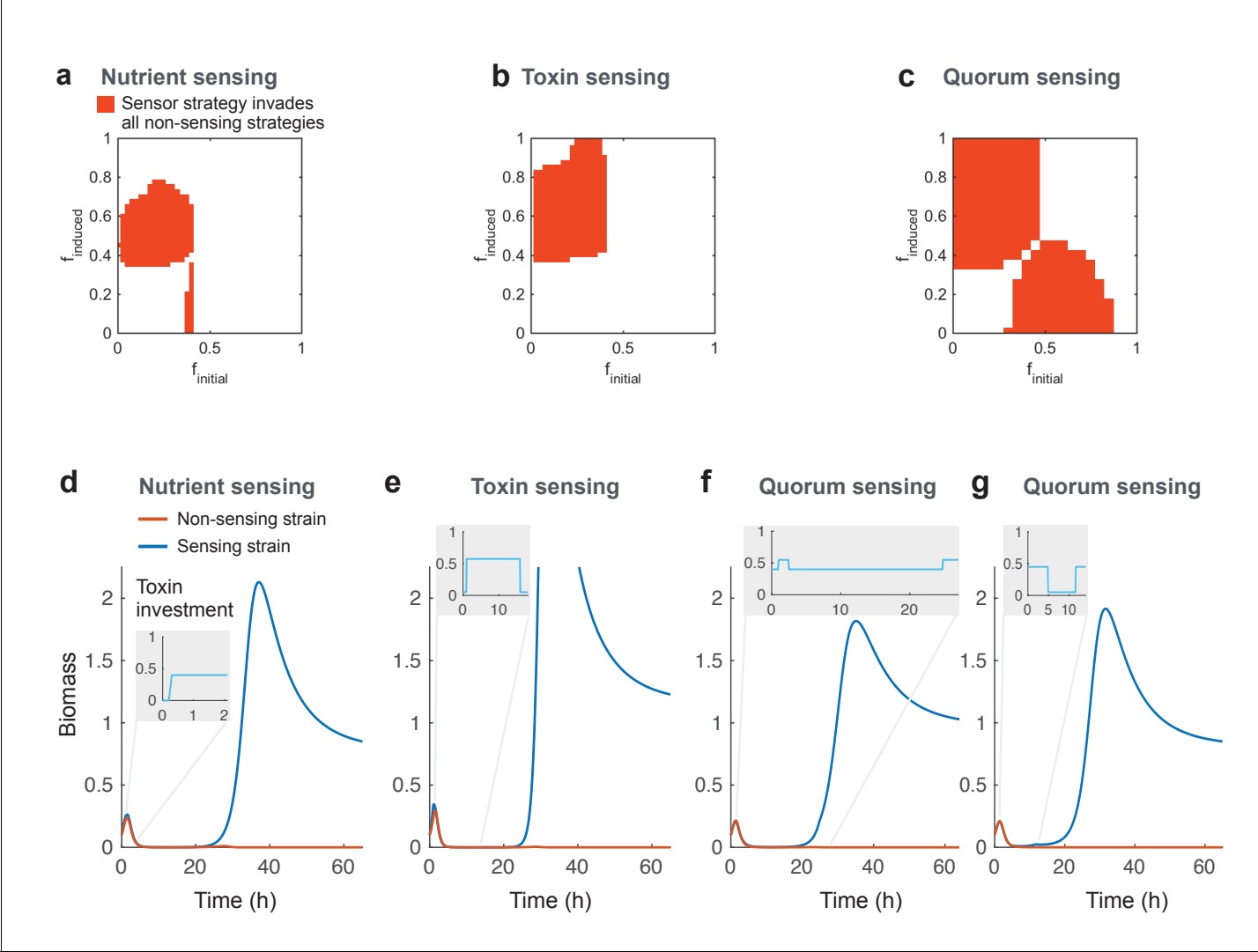

**Figure 2.** Regulated toxin production outcompetes and evolutionarily replaces constitutive toxin production. Using a deterministic grid search, we find nutrient-sensing, toxin-sensing, and quorum-sensing strategies that can stably invade the entire range of non-regulated producer strategies (a-c, red areas). In these plots, the effects of two parameters on competitive outcome are shown: $f_{initial}$, the toxin investment of a sensing strain at the initial state, and $f_{induced}$, the toxin investment after the signal passes a certain threshold. Red areas indicate combinations of $f_{initial}$ and $f_{induced}$ where at least one threshold value allows stable invasion. Illustrative competitive dynamics are shown for the optimal non-sensing strategy against (d) nutrient-sensing, (e) toxin-sensing, and (f) quorum-sensing (upregulates toxins at high quorum) and (g) quorum-sensing (downregulates toxins at high quorum). Grey insets show investment in toxin production as a function of time. Regulation allows tactics that use toxins more efficiently and effectively than constitutive producers. All parameters take standard values as given in *Table 1*.

higher initial investment in toxin production in order to ensure that enough toxin is made in the time that a strain has to compete (*Appendix 1—figure 5*).

How do the different regulatory strategies achieve their success? For the great majority of cases, successful strains evolve to upregulate their attack after a delay, either based on the detection of low nutrients, high quorum, or high levels of the competitor's toxin (*Figure 2d–f*). In some cases, there is no toxin production before this upregulation, as in the canonical model of quorum sensing that turns a trait from off to on. In other cases, the strategy that evolves is to begin with a baseline of constitutive production before upregulating this further upon activation (*Figure 2a–c*, with example shown in *Figure 2f*), something also seen in real systems (*Mavridou et al., 2018*). A difference between nutrient and quorum sensing versus toxin-based regulation is that examples of the latter

not only upregulate toxin production after a delay, they also downregulate the toxin if the competitor is killed off (*Figure 2e*).

We also discovered winning strategies that function by *down*regulating toxin production after a delay. For nutrient-based regulation, there is a narrow parameter range (the small vertical strip in the lower part of *Figure 2a*) where strategies begin aggressively with the expression of toxin and then downregulate it when nutrients are limited (dynamics not shown). For quorum-sensing strategies, some also start with high toxin investment, but these strategies are more complex. These downregulate toxins and invest in growth once they reach a high density, but will reactivate it again if their cell numbers drop due to toxin attack (*Figure 2g*).

In each case, regulated strategies win by only making high levels of toxin at certain times, thereby saving energy relative to constitutive producers. A corollary is that, if toxin production is cost free, regulation will no longer be benefitical relative to constitutive production. But, assuming that there is some costs to toxin production, regulation is expected to be favoured by natural selection.

In sum, there are regulated strategies of each of the three types under study that can evolutionarily replace all non-regulated strategies. However, this analysis is based on regulated strains invading metapopulations consisting of a single constitutive strategy. In some contexts, a focal strain may face a variety of competitors. Consider, for example, a situation where migration brings in a range of competing species, each optimised to a different environment. To consider this scenario, we next ask how the different sensing strategies fare in competition with a standing diversity of constitutive strategies. We introduce diversity by letting the different sensory strategies (i.e. nutrient sensing, toxin sensing, and quorum sensing) face an increasingly diverse mix of constitutive toxin-producing opponents. We assume that the standing diversity of constitutive producers is not itself affected by the evolution of the regulated startegies, that is, there is no coevolution (we consider coevolution in the next section, however). For each set of opponents, therefore, we can identify the best performing regulated strategies simply as those that obtain the highest average biomass across the competitions with the set of opponents (Materials and methods). Based on the simulated data, we also fitted a linear regression model with sensing type as a categorical predictor and number of competitors a numerical predictor (see Materials and methods).

When opponents have a single strategy (lowest diversity), the toxin sensing strategy is most efficient in terms of its final biomass produced (*Figure 3a*, left panel). Moreover, the toxin sensing strategy deals most effectively with diverse competitors (*Figure 3a*) with the regression analysis showing a 2.5 times higher fitness for toxin sensing relative to the other strategies (p-value < 0.001). The success of the toxin sensing strategy is associated with the reliable activation of toxin production when sensing another toxin. Quorum sensing also activates toxin production during the competition but, in some cases, is defeated without being able to attack back. This gives rise to the observed bimodal outcome of the quorum-sensing strategy (*Figure 3a*). The nutrient-sensing strategy, by contrast, attacks first and then deactivates later when nutrients decrease.

This superiority of toxin sensing is robust across a range of parameters, including different toxin efficiencies, toxin loss rates, and nutrient concentrations (*Appendix 1—figure 7*). There is a clear post hoc intuition to this result. A strain that only engages in conflict when attacked will be best able to deal with a range of strategies that differ in their propensity and ability to attack. More specifically, as seen in the last section, these strains inactivate toxin production after a weak opponent is eliminated, thus employing the toxin efficiently. We can directly demonstrate the importance of this tactic of toxin inactivation by shortening the duration of the strain competitions such that toxin-sensing strains do not have the opportunity to downregulate toxin production. For short competition times, while regulated strategies still outperform unregulated ones (*Appendix 1—figure 5*), the toxin sensing strategy fails to evolve a superior performance over the other modes of regulation (*Figure 3b*).

## The coevolution of regulated attack strategies

We have considered how regulated attack strategies perform in the face of constitutive strategies that vary in their level of aggression, and in the face of varying levels of diversity in these opponents. This revealed that regulation is generally beneficial and indicated that the sensing of an opponent's toxin is often the best performing strategy. However, this analysis is artificial in the sense that bacteria with regulated strategies are also likely to compete against one another. Therefore, we next ask, which sensing strategy is most successful when coevolving with other sensing strategies? We first

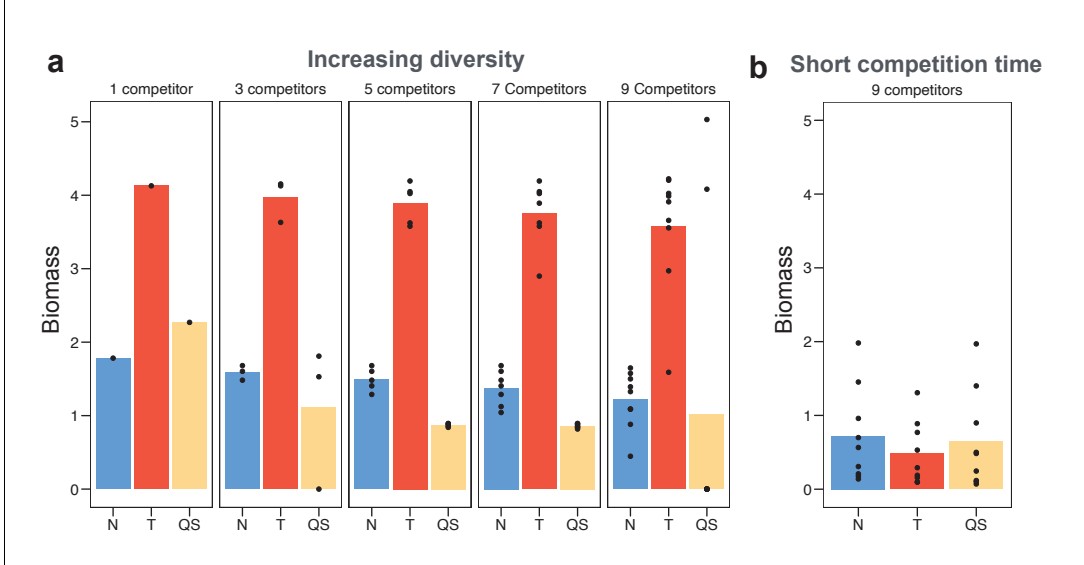

**Figure 3.** Toxin sensing is the most versatile strategy against a range of different competitor strategies. (a) We optimised (using a grid search) each of the sensing strategies first against a single constitutive producer (left-hand side panel) and then against an increasing diversity of producers (other panels). As an example, the nine competitors (right-hand side panel) have toxin investment $f$=0.1,0. 2,…,0.9 and we optimise each of the three sensing strategies in terms of the sum of their final biomasses across all nine competitions. We show the final biomasses of individual competitions as points and the average biomass as bars. The toxin-sensing strain (red coloured bars) performs best, both against the single strategy and against mixtures of strategies. Among the other two sensory strategies, quorum sensing (yellow) has a higher variation of biomass than nutrient sensing (blue) across individual fights. The benefit of sensing toxin is robust for diverse environmental conditions (**Appendix 1—figure 7**). (b) Shortening the competition time ($t_{end}$ = 6 hr) removes the benefit of toxin sensing. When not mentioned, parameters take the standard values as given in **Table 1**.

consider strains that interact with others that have a similar attack strategy, regulated by the same environmental cue. For each of the three types of regulation, we then search for the optimal strategy using a genetic algorithm (see Materials and methods) (**Figure 4a**). Following the logic of the earlier models, the optimal strategy is defined as one that will, on average, obtain the highest biomass across competitions with all other possible strategies.

When competing with the same strategy, all strategies initially evolve to increase toxin production during the competition ($f_{initial} < f_{induced}$) (**Figure 4b**). More specifically, strains responding to nutrient depletion initially produce near zero toxins ($f_{initial}$ = 0.05) until they activate toxin investment, at a level higher than the optimal fixed investment (evolved $f_{induced}$ = 0.50, while $f^*$ = 0.35). In comparison, strains responding to quorum sensing invest in more toxin initially ($f_{initial}$ = 0.11) and also more when activated ($f_{induced}$ = 0.60). The quorum-sensing strategy is expected to be able to afford to invest more in toxin production because, unlike nutrient sensing, strains can reduce toxin investment again if biomass drops too low, thereby saving energy. The toxin-sensing strategy is different again. It invests near zero toxin at the start of the competition ($f_{initial}$ = 0.01) and responds very strongly if a competitor attacks ($f_{induced}$ = 0.73). Interestingly, the corollary is that, at evolutionary equilbrium (when it will meet an identical toxin strategy), both remain passive and achieve a high biomass (**Figure 4b** center). This outcome has similarities to the success of 'tit-for-tat', a reciprocal cooperating strategy in the classic evolutionary game theory tournament of **Axelrod and Hamilton, 1981**. There, tit-for-tat succeeds by benefiting from mutual cooperation whenever others cooperate, while maintaining the ability to shut off cooperation whenever it meets a non-cooperative strategy. When this success leads to all individuals playing tit-for-tat, the result can be that all interactions end up as cooperative, akin to the emergence of a peaceful productive strategy in our model.

The evolution of a peaceful outcome is specific to the ability to reciprocate; we do not observe it for nutrient or quorum sensing. Nevertheless, we have identified a route by which bacteria might evolve the peaceful resolutions seen in animal and human conflicts (**Axelrod and Hamilton, 1981**; **Kokko, 2013**; **Freedman, 1989**). However, the model assumes that strains will only interact with other genotypes that are adopting similar strategies for warfare. This is far from guaranteed in

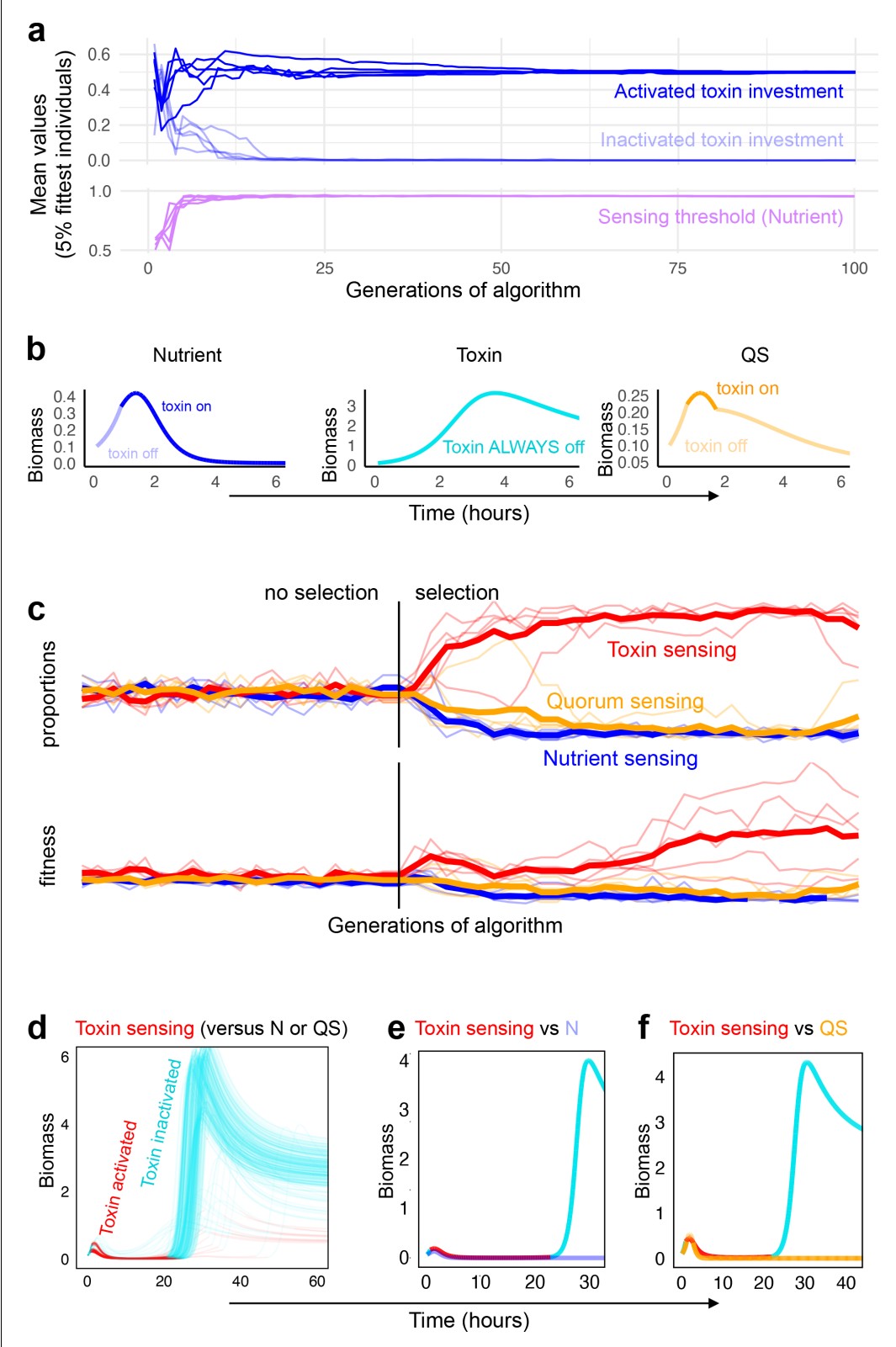

**Figure 4.** Coevolution of sensing strategies. Panels **a** and **b** follow the evolution of regulation when they compete solely against strains with the same type of regulation, for example, quorum sensing versus quorum sensing; Panels **c–f** follow evolution when all three types of regulation compete together. (a) Representative example of evolutionary convergence of the sensing parameters for the case of nutrient-sensing strategies competing solely against other nutrient-sensing strains. Displayed is the mean parameter value of the 5% fittest strains at each generation for 10 independent runs

*Figure 4 continued on next page*

*Figure 4 continued*

of the algorithm. (**b**) The biomass dynamics of the three evolutionarily stable sensing strategies at equilibrium (when in competition with a strain with an identical strategy). Areas where the dynamics are shown in a pale tone indicate time intervals where toxin was downregulated. For toxin sensing, toxin production remains deactivated thoughout. (**c**) Evolution in tournaments where all strategies compete against one another, showing the dominance of the toxin sensing strategy. For five independent runs of the tournament the upper panel shows population fraction of the different strategies, the lower panel shows individual fitness averaged per sensing type. The thick lines give the average across runs. The tournament starts with 20 generations (left of the vertical line) without selection, after that, strategies are selected based on their competitive fitness. (**d**) Winning strategies: shown is the range of dynamics of the optimised toxin sensing strategies against quorum sensing and nutrient-sensing strategies that evolved in one realisation of the tournament. Red and turquoise indicate activated and inactivated toxin production, respectively. (**e**) Example of a competition between one of the winning toxin-sensing strains meeting a nutrient-sensing strain evolved in the tournament. Red and turquoise, respectively, indicate upregulated and downregulated toxin production for the toxin-sensing strain, other strain is shown in light blue. (**f**) Example of a competition between one of the winning toxin-sensing strains and a quorum-sensing strain evolved in the tournament. Red and turquoise, respectively, indicate upregulated and downregulated toxin production for the toxin-sensing strain, other strain is shown in yellow. All parameters take standard values as given in *Table 1*.

bacteria as there exists considerable variability in weapons and their regulation, even within a single species (*Mavridou et al., 2018*). Moreover, microbial communities typically contain many strains and species, suggesting again that a given strain has the potential to meet a diversity of competitors and strategies.

We therefore sought to capture this complexity with a final model in which all possible regulated strategies are able to compete against each other, again using a genetic algorithm to identify optimal strategies (see Materials and methods). Despite a great number of potential combinations (over two million different competitions), and with different sets of hyperparameters of the genetic algorithm, we again see a clear winner in toxin-based regulation, both for our normal parameters (*Table 1*, *Figure 4c*) and for sweeps that consider broad ranges of these parameters (*Appendix 1—figure 8*) and a wide range of initial frequencies of the two strains (*Appendix 1—figure 9*). Moreover, as for competition against unregulated strategies (*Figure 3*), the success of toxin-based regulation in contests with other strategies does not come from an ability to avoid conflict and create peaceful outcomes. Instead, the winning strategies are typically aggressive when they meet another strain and they only downregulate their toxins once an opponent is on its way to being eliminated (*Figure 4d–f*). And, as for competition against unregulated strategies, this ability to become passive is key to their success. For short competitions, there is no benefit in turning off an attack and the competitive benefit of reciprocity over other regulated strategies is lost (*Appendix 1—figure 10*).

## Conclusions

Bacteria use a wide variety of weaponry to harm other strains and species, which is typically under tight regulation (*Ghequire and De Mot, 2014*; *Granato et al., 2019*; *Stein, 2005*; *Michel-Briand and Baysse, 2002*; *Cascales et al., 2007*). How bacteria employ these mechanisms of attack is central to understanding why a particular species or pathogen can invade and persist in communities, while others cannot (*Granato et al., 2019*; *Kommineni et al., 2015*). Here, we have explored the evolutionary logic underlying strategies of bacterial attack. We find that toxin production is favoured under many conditions, particularly when toxins are effective and long-lasting and when the potential for population expansion is limited (*Table 1*). The prevalence of aggressive strategies in our model is consistent with the widespread use of toxins by bacteria (*Granato et al., 2019*), and the associated intensity of competition observed in experiments (*Chao and Levin, 1981*; *Mavridou et al., 2018*; *Oliveira et al., 2015*). We also find that well-regulated attacks can consistently outcompete strategies that lack regulation (*Figure 2*). This is because the benefit of employing a toxin not only changes with different competitors but also within a single competition over time. Regulation allows a strain to better tune its behaviour and follow the optimal investment at any given situation. However, the three major classes of bacterial regulatory network are not always equivalent ways to control attacks. Across a diverse range of potential competitors, responding directly to incoming attacks is the most robustly successful strategy (*Figures 3,4*).

Our modelling implicitly captures spatial structure at the metapopulation level with discrete patches of bacteria that compete with each other. Within patches, our ODE model best reflects environments with limited spatial structure where cells of different genotypes are mixed together. However, bacteria do also display fine scale spatiogenetic structuring within their communities

(*Nadell et al., 2016*; *Stacy et al., 2016*; *Krishna Kumar et al., 2021*). Here, our model has the potential to capture the outcome of competition at the interface of two strains, which is expected to be critical for success and persistence in such communities (*Granato et al., 2019*). However, there is clear potential for other effects of local spatial structure on sensing strategies that we do not capture. For example, in contrast to the detection of competitor's toxins, responses to quorum sensing and nutrient depletion may occur first in the middle of a patch of cells, where toxin production has the least benefit as toxin receivers are mainly clone-mates (*Inglis et al., 2009*; *Wechsler et al., 2019*).

Our work predicts that sensing incoming attacks through direct or indirect means should be a widespread way of regulating toxins and other modes of attack. This hypothesis lends itself to empirical testing via the study of bacterial behaviour during toxin-mediated competition with other strains and species. Some examples of reciprocation already exist. Many bacteria upregulate attack mechanisms via stress responses that detect cell damage (*Cornforth and Foster, 2013*). This includes recent evidence of reciprocation between warring *Escherichia coli* strains where DNase protein toxins activate toxin production in competing strains via the SOS response to DNA damage (*Mavridou et al., 2018*; *Krishna Kumar et al., 2021*; *Gonzalez et al., 2018*; *Granato et al., 2019*). Because many antimicrobials target the DNA of cells (*Janion, 2008*; *Gillor et al., 2008*), sensing DNA damage is likely to be a relatively robust way to achieve reciprocity. But there are other mechanisms; *Pseudomonas aeruginosa* senses incoming attacks via the type six secretion system (T6SS) of competitors, which delivers toxin via the molecular equivalent of a speargun (*Basler and Mekalanos, 2012*; *Basler et al., 2013*). Upon detecting an incoming attack, a cell will activate its own T6SS in response (*Basler et al., 2013*). Consistent with our findings, recent work suggests that a key benefit to reciprocation via the T6SS is the ability to save energy and only attack when necessary, alongside a benefit that comes from improved aiming which is specific to this mode of attack (*Smith et al., 2020*). Finally, there is evidence that bacteria may also detect and respond to incoming attacks via proxies such as the detection of lysate produced when surrounding cells are killed (*LeRoux et al., 2015a*), or molecules that are made by an attacker alongside a toxin (*Cornforth and Foster, 2013*; *LeRoux et al., 2015b*).

There is also evidence that bacterial toxins can be regulated via nutrient depletion and quorum sensing (*Ghequire and De Mot, 2014*; *Cascales et al., 2007*). Our models of regulation by quorum or nutrients typically predict that attacks will evolve to be activated at high quorum or limited nutrients, which recapitulates the typical directionality of the regulation observed in nature (*Chandler et al., 2012*; *Fontaine et al., 2007*; *Inaoka et al., 2003*). However, if detecting damage is the best basis for attack, why do some bacteria use these other forms of regulation? For short competition times, our model predicts that the three regulatory strategies are largely equivalent (*Figure 3* and *Appendix 1—figure 10*). A short duration of competition between strains removes the benefit of decreasing toxin production once an attacker has been overcome. Under these conditions, the evolutionary path to one form of regulation may largely be determined by differences in costs for regulatory networks and which pre-existing regulatory systems are available for co-option (*Cotter and DiRita, 2000*; *Hockett et al., 2015*). We predict, therefore, that mechanisms to reciprocate attacks are particularly valuable in environments where warfare commonly leaves a victor unchallenged for a long time afterwards. Consistent with this, one of the clearest examples of reciprocation occurs in *E. coli* (*Mavridou et al., 2018*; *Krishna Kumar et al., 2021*; *Granato et al., 2019*), which uses colicin toxins to displace other strains and persists for long periods within the mammalian microbiome (*Gillor et al., 2009*).

Another possible explanation for why some bacteria do not use cell damage to regulate their toxins comes from the notion of 'silent' toxins. These are toxins that are not easily detected by the cell's stress responses, which may limit the potential for a toxin-mediated response. For example, some toxins depolarise membranes (*Yang and Konisky, 1984*) and may be favoured by natural selection specifically because they do not provoke dangerous reciprocation in competitors (*Gonzalez et al., 2018*). In other cases, bacteria appear to use multiple forms of regulation in order to integrate information from multiple sources (*Cornforth and Foster, 2013*). For example, *Streptomyces coelicolor* regulates antibiotic production via both nutrient limitation (*Hesketh et al., 2007*) and mechanisms that detect incoming antibiotics (envelope stress [*Hesketh et al., 2011*]). A potential future use of our modelling framework would be to study how these combined regulatory strategies evolve.

Bacteria use diverse regulatory networks to attack and overcome competitors, and there is much still to understand about their evolution. Here, we have identified general principles for the function of these networks in bacterial warfare. We find there are great benefits using regulation to time an attack; both to minimise its cost and maximise its effect on an opponent. We also find that regulation that enables reciprocation can be particularly beneficial. If cells only attack when attacked, they invest their energy where and when it is most needed: against aggressive opponents. Our findings are mirrored in the classical predictions from the game theory of animal combat, which suggested that adopting a reciprocal and retaliatory strategy can be effective (*Maynard Smith and Price, 1973*; *Kokko, 2013*; *Freedman, 1989*; *Enquist and Leimar, 1990*). However, the predicted outcome was typically one of peace and the avoidance of conflict, which is indeed what is observed in many animal contests (*Briffa M, 2013*). In contrast to such lessons, experimental work suggests that bacteria often engage in deadly conflict (*Abrudan et al., 2015*; *Mavridou et al., 2018*; *Oliveira et al., 2015*; *Gonzalez et al., 2018*; *Be'er et al., 2009*; *Vetsigian et al., 2011*). Our models offer an evolutionary rational for this observation. The regulation of combat in bacteria is not usually about avoiding conflict; it is about timing an attack and downregulating it once a competitor is no longer a threat.

## Materials and methods

### Overview

In this study we use a modelling framework that captures two scales of competition (*Figure 1*). At the local level, we model bacterial strain competitions using systems of ODEs. These equations are well suited to model temporal dynamics on the relatively short ecological timescales at which bacterial strains interact with nutrients and competitors. At the global level, we model the evolution of different strategies within a metapopulation. This metapopulation level allows us to follow the evolution of different strategies across much longer *evolutionary* timescales, and to capture the important interplay of local and global fitness (*Figure 1d*). We use this game theory framework to identify strategies that are evolutionarily successful against a diversity of possible competitors. All questions addressed in this work require both layers of modelling. The system of ODEs that models constitutive toxin production is described in the next section and forms the basis for all of the models. Evolution at the metapopulation level is implemented using a common logic (*Figure 1c,d*), using variations that capture a range of questions and evolutionary scenarios as detailed below.

### A differential equation model of bacterial warfare

Our model captures pairwise competitions between bacterial strains, which have the potential to produce toxins (*Figure 1a*). This first model allows a strain to have a fixed investment into its toxin – below we describe the extension of this model that allows toxin regulation in response to external cues. We employ ODEs, which are well suited to capture the temporal dynamics of strain interactions happening at ecological timescales. A number of different models have been used to study the evolution of bacterial public good regulation (*Niehus et al., 2017*; *Heilmann et al., 2015*; *Kümmerli and Brown, 2010*). Here, we follow *Bucci et al., 2011*, because they model both nutrients and toxins explicitly, which are both important cues for the regulation of toxin production. We study a competition between two strains that each possess a toxin that does not harm the cells of the producer strain, but does harm the other strain. In reality, strains may carry multiple toxins and resistances (*Cordero et al., 2012*; *Gordon et al., 1998*) and the evolution of multiple mechanisms of attack and defence is an interesting question in its own right. However, we focus here on a single toxin produced by each strain. We also describe the dynamics of the nutrients and cell densities in a well-mixed environment. The interactions of cells, nutrients, and toxins can be described by the system of ODEs:

$$\frac{dC_A(t)}{dt} = (1-f_A)\mu_{max}\frac{N(t)}{N(t)+K_N}C_A(t)-kT_B(t)C_A(t),$$

$$\frac{dC_B(t)}{dt} = (1-f_B)\mu_{max}\frac{N(t)}{N(t)+K_N}C_B(t)-kT_A(t)C_B(t),$$

$$\frac{dT_A(t)}{dt} = f_A\frac{N(t)}{N(t)+K_N}C_A(t)-l_TT_A(t),$$

$$\frac{dT_B(t)}{dt} = f_B\frac{N(t)}{N(t)+K_N}C_B(t)-l_TT_B(t),$$

$$\frac{dN(t)}{dt} = -\frac{N(t)}{N(t)+K_N}(C_A(t)+C_B(t)),$$

$$(1)$$

where $C_A(t)$ and $C_B(t)$ denote the biomasses of cell strains $A$ and $B$, respectively, $T_A(t)$ and $T_B(t)$ denote the biomass of each strain's toxin, and $N(t)$ denotes the concentration of a growth-limiting nutrient for which both strains compete. We consider a pool of nutrient that is depleted by the cells. Similarly to *Nadell et al., 2008*, we describe the energy that is available to the cells by the Monod equation, in which $K_N$ is the nutrient saturation constant. The maximum growth rate is given by $\mu_{max}$. Toxins kill with efficiency $k$ and are lost with rate $l_T$. We assume that all toxins have identical loss and killing rates in order to remove biochemical differences between strains and focus our analysis on the effects of different production strategies.

For constitutive toxin production, the strategy of a strain is given simply by a fixed $f$ ($f \in [0,1]$), which captures the investment into toxin production relative to cell biomass. The production of anti-biotics and bacteriocins can have significant metabolic costs and can even require a cell to lyse, as occurs with colicins and pyocins (*Cascales et al., 2007*; *Nakayama et al., 2000*). We model the cost of toxin production on cellular growth as a linear allocative trade-off function in the growth term (*Bucci et al., 2011*). For example, a strain that invests $f = 0.1$ into its toxin will only reach 90% of its maximal growth rate.

The dynamics of cells, nutrients, and toxins are modelled as continuous for their typical range. But when a cell strain reaches a very low concentration ($C(t)=10^{-6}$), we assume that stochastic extinction occurs such that cell concentration drops to 0. Further, our model assumes a limited lifetime of the local patches by stopping the dynamics when 24 hr (or less for the analysis of shortened competition times, *Appendix 1—figure 5*) have passed.

We solve the system of ODEs numerically using an implicit Euler method. This numerical scheme is implemented in MATLAB (version 9.5.0.944444) (*MATLAB, 2018*). Our implementation solves the equations (*Equation (1)*) until the defined end time. We avoid numerical issues due to negative state variables by setting any state variables reaching a value below $10^{-8}$ to 0.

## A model of regulated toxin attack

To extend the above model to include sensing, toxin production of bacterial strain $A$ is either a function of nutrient depletion, toxin of strain $B$, or of quorum sensing (given as cell biomass of strain $A$). Each signal triggers toxin production via a simple on-and-off switch (*Cornforth and Foster, 2013*) so that the toxin production of strain $A$ is given through one of the equations:

$$f_A = f_{initial} + (f_{induced}-f_{initial})H[(N(t=0)-N(t))-U_N],$$

$$(2)$$

$$f_A = f_{initial} + (f_{induced}-f_{initial})H(T_B(t)-U_{TB}),$$

$$(3)$$

$$f_A = f_{initial} + (f_{induced}-f_{initial})H(C_A(t)-U_{QS}),$$

$$(4)$$

where $H$ is the Heaviside step function given as

$$H(x) = \begin{cases} 0, x<0 \\ 1, x \geq 0 \end{cases}$$

$$(5)$$

and where

$$f_{initial} \in [0,1] \text{ and } f_{induced} \in [0,1].$$

These equations of regulated toxin production each comprise the initial investment into toxins ($f_{initial}$) when the trigger term is deactivated and the trigger term itself. The trigger term contains a Heaviside step function and becomes active when the signal increases over the sensing threshold ($U_N$/$U_{TB}$/$U_{QS}$). When activated, the trigger term changes the initial toxin investment ($f_{initial}$) to become the induced toxin investment ($f_{induced}$). We allow the induced toxin investment to be smaller (when the signal is a repressor) or larger than the initial toxin investment (when the signal is an activator).

## Invasion analysis

We use our first models to predict the optimal constitutive toxin production strategy across different ecological conditions. Here, the assumed scenario is a monomorphic metapopulation (all strains have identical warfare strategy), where a rare mutant strategy appears that may or may not invade this metapopulation. As time progresses toward infinity, the metapopulation will finally be dominated by a strategy that can invade the metapopulation of any other strategy and that can itself not be invaded. We implement this scenario using classic pairwise invasion analysis. More specifically, we employ game theory and, in particular, invasion analysis to find the best strategies (*Nowak and Sigmund, 2004*; *McElreath and Boyd, 2013*), where the best strategy is one that, if adopted by the whole population, cannot be invaded by any other strategy. These strategies are also called evolutionarily stable strategies (*Maynard Smith, 1982*).

We follow previous work (*Oliveira et al., 2014*; *Cremer et al., 2012*) by assuming a microbial life cycle that consists of a seeding step where local patches are seeded with two competing strains, a competition step where strains grow and interact according to the differential equations explained above, and a mixing step where cells from all patches disperse and mix, leading to a new seeding episode (*Figure 1c*). The proportion of the different strains (or strategies) that are seeded is determined by the strain frequencies after the competition step. Without explicitly modelling this life cycle, invasion analysis (*McElreath and Boyd, 2013*) asks whether a particular strain with strategy $f_{inv}$ when rare can invade a population dominated by another strategy $f_{res}$ (the resident *Weibull, 1997*; *Figure 1d*). To answer this, we calculate the fitness of the resident strategy ($w_{res}$) and the fitness of the invading strategy ($w_{inv}$). The fitness of the resident is its final biomass when in competition with an identical strategy so that $w_{res} = w(f_{res}|f_{res})$ and the fitness of the rare invader is determined by its final biomass in the competition between invader and resident strategy, $w_{inv} = w(f_{inv}|f_{res})$. We then calculate the invasion index for an invading strategy according to *Mitri et al., 2011* as

$$I_{inv} = \frac{w_{inv}}{w_{res}} = \frac{w(f_{inv}|f_{res})}{w(f_{res}|f_{res})}. \tag{6}$$

When the invasion index $I_{inv}$ is larger than 1, the rare strategy can invade the resident strategy; when the index is smaller than 1, the rare strategy cannot invade, and it disappears. Finally, we also test for back-invasion and compute $I_{inv}$ for when the resident is rare and the mutant is the resident. We implement strain competitions by solving the system of ODEs described above. We define evoluationarily stable strategies as those strategies that have an $I_{inv}$ larger than 1 against all studied competitors (and both as rare and resident strategy). By calculating the invasion index for a large number of invading strategy-resident strategy pairs, we obtain a pairwise invasibility plot (*Brännström et al., 2013*) (insets in *Appendix 1—figure 4*). Using this plot, we find a single evolutionarily stable strategy *f\** that can invade all strategies and that cannot be invaded by any other strategy. We determine this globally optimal strategy using the algorithm outlined in the Appendix 1—code 1. We can then ask how the parameters of the model affect the evolution of toxin investment (*Table 1*).

## Invasion analysis of sensing strategies

We next ask whether regulated strategies will evolutionarily replace constitutive production. Here, the ecological scenario is the same as above: monomorphic populations of constitutive toxin production strategies are threatened to be invaded by rare strategies that can sense (*Figure 1d*). We perform a parameter grid search that tests a large number of sensing strategies (stepping: $\Delta f_{initial}/\Delta f_{induced} = 0.02$ and $\Delta U = 0.002$, constraints: $f_{initial} \in [0, 1]$, $f_{induced} \epsilon [0, 1]$, $U_N \in [0, 1]$, $U_{TB} \in [0, 20]$, $U_{QS} \in [0, 20]$) against the range of constitutive strategies. For the constitutive strategies, we select

from a fine grid spacing that also includes the optimal constitutive strategy ($f_{fixed}$ = [0.00, 0.01, 0.02, ..., 1.00]). For each pair of sensing and non-sensing strategies, we compute the invasion index once for the sensing strategy as the resident and again for the non-sensing strategy being the resident. We then search for those sensing strategies that can invade all non-sensing strategies and that themselves cannot be invaded by any other non-sensing strategy. We show where those strategies lie in the parameter space of $f_{initial}$ and $f_{induced}$ (**Figure 2a-c**).

## Sensing strategies against standing diversity

We also study the evolutionary success of the three different types of sensing when being in constant competition with a diverse set of competitors. Here, the ecological scenario is a polymorphic metapopulation – a mix of different constitutive production strategies – with a given diversity. We assume that this diverse set of strategies is not influenced by evolution in the focal sensing strategy due to, for example, immigration that continually resupplies the diversity of competitors. We then ask what happens when a rare sensing strain enters this metapopulation, where its success depends on its success across pairwise competitions with the different resident strategies.

We implement this by competing focal sensing strategies against a set of different constitutive strategies and computing their fitness from the average biomass produced across those competitions. Specifically, for each of the three different sensing types, we perform a parameter grid search, creating a large number of predefined strategies across the parameter range of $f_{initial}$ ($\in$[0,1], at increments of 0.05), $f_{induced}$ ($\in$[0,1], 0.05) and respective thresholds $U_N$ ($\in$[0,1], 0.02), $U_{TB}$ ($\in$[0.001,4], 0.0005), and $U_{QS}$ ($\in$[0.01,1.2], 0.01). Each of those sensing strategies is competed against a fixed set of constitutive strategies, one at a time, by solving the above system of equations. We then compute for each sensing strategy the average fitness across its competitions. Within each of the three sensing types we find the single strategy with the highest average fitness. For those winners we show the average fitness as bars in **Figure 3** together with the fitnesses obtained against each individual constitutive strategy. We repeat this entire procedure for five different levels of diversity among the constitutive strategies. Starting with the lowest diversity set, which contains only a single constitutive strategy ($f$ = 0.5), we then add increasingly extreme strategies, yielding three competitors ($f$ = 0.4,0.5,0.6), five competitors ($f$ = 0.3,0.4,0.5,0.6,0.7), seven competitors ($f$ = 0.2,0.3,0.4,0.5,0.6,0.7,0.8), and finally nine competitors ($f$ = 0.1,0.2,0.3,0.4,0.5,0.6,0.7,0.8,0.9).

Using the simulated results, we fit linear regression model with the sensing type as a categorical predictor variable and the number of competitors as a numerical predictor variable. The regression takes the form

$$F_i \sim N\left(\alpha_{S[i]} + \beta D[i], \sigma\right)$$

where $F_i$ is the fitness of the $i$th competition, which we assume to be normally distributed around a mean given by a linear equation and with standard deviation $\sigma$. The fixed intercept is given through $\alpha_{S[i]}$, where $S[i]$ is the $i$th element of integer vector $S$ that contains only two possible values indicating whether the toxin sensor or a different sensor is in the competition $i$. $D[i]$ is the $i$th element of integer vector $D$, which gives the number of competitors in the $i$th competition. Finally, $\beta$ gives the change of fitness when adding one competitor. We fit the regression model using R (version 3.6.1) (**R Development Core Team, 2011**).

## Genetic algorithm

Finally, we study which sensing strategies are most successful in competition with other sensing strategies. We study this both within each type of sensing and across all three different types. Here, the scenario is a polymorphic metapopulation of coevolving sensing strategies. Mutation and migration create new strategies inside this population. A strategy's achieved biomass in pairwise competitions with other strains determines its ability to stay and amplify in the metapopulation. The model initially studies a wide variety of strategies competing with one another. However, as time passes, the metapopulation converges and consists of increasingly optimal strategies. As this happens, the analysis then approximates the invasion analyses described above, where most strains are largely identical and rare mutants are pitted against this majority in the metapopulation (**Figure 1d**).

Specifically, we use a genetic algorithm to search for the evolutionarily stable strategy in the large space of possible strategies of a single type of regulation (and also in the space of all possible

regulating strategies). This algorithm adapts the typical structure of a genetic algorithm (*Melanie, 1996*) where in each round a population of individuals is first tested to evaluate fitness and it is then replaced by a new daughter generation. Individuals of this new generation are created by a mix of cloning and mutating individuals from the previous parent generation selected based on their fitness and by addition of novel random strategies. As is typical in non-adaptive algorithms , the control parameters of the algorithm (e.g. number of generations, number of strategies in the population, rate of mutation, etc.) are chosen to achieve short simulation times and good convergence behaviour as determined by visually inspecting the distribution of population parameters over time (*Melanie, 1996*). Our population of competing strategies has a constant size of *n*=60. Initially a set of random strategies is created, whereby the three parameters that define an individual sensing strategy are drawn from a uniform distribution with given parameter constraints ($f_{initial} \in [0, 1]$, $f_{induced} \in [0, 1]$, $U_N \in [0, 1]$, $U_{TB} \in [0, 4]$, $U_{QS} \in [0, 1.2]$). The constraints for the sensing thresholds take the range of the respective signals as they are observed across the large number of competitions performed in the invasion analysis of sensing strains described above. (For the initial population in the case where all three sensing types compete, the sensing type is chosen at random with equal probability for all three types and, to avoid long run times and artefactual superiority due to parameter constraints, initial parameter values start at the optimum from the within strategy competition.)

In every round then, each strategy competes against all *n* strategies, including its own type. The final biomass of every strategy is summed across its competitions to give its competitive fitness. Then, a new daughter generation is generated. The four most competitive parent strategies are chosen to move into the next generation without parameter mutation, 36 strategies are drawn from the parent generation with probability proportional to their fitness and one of their parameters is mutated by adding a value drawn from a normal distribution with mean of 0 and standard deviation of 0.001. If, after mutation, a daughter strategy violates the parameter constraints, the random draw gets repeated until the constraints are met. Finally, 10 immigrant strategies are generated by choosing their sensing parameters as random draws from a uniform distribution within the constraints. (In the case of all three strategies competing, the sensing type is first drawn at random with equal probability, and then the sensing parameters are drawn at random.) For the competition between types of a single sensing strategies, the algorithm is run for 100 generation. (For the tournament with all three strategies, we ran the first 20 generations without selection, where we replace the population each generation with migrants, to allow comparison with the case where selection occurs, *Figure 4c*). The evolving parameter values for the top four strategies are averaged for each generation and saved (*Figure 4a*). The averaged values in the last timestep give the evolutionary stable strategy for each tournament (*Figure 4b*). In our sensitivity analysis, we also examine the results of the genetic algorithm with alternative sets of control parameters, including a smaller and a larger size of the mutation standard deviation (0.01 and 0.0005), a smaller and larger proportion of 'migrating' strategies in each generation (5 of 60, and 20 of 60), and five different sizes of the population of strategies (50, 70, 80, 90, 100). This yields a total of 20 alternative parameter combinations.

## Code availability

The MATLAB code for the regulated toxin model, the invasion analysis, and the evolutionary tournament is available on GitHub (https://github.com/reneniehus/bact_warfare, copy archived at swh:1:rev:923e104aa634230547ba464c6bc8fee07f662ffa, *Niehus, 2021*).

## Acknowledgements

Thank you to Michael Bentley, Jonas Schluter, Kat Coyte, Nicholas Davies, Wook Kim, and Rolf Kümmerli for feedback. RN is funded by the EPSRC-funded Systems Biology Doctoral Training Centre studentship EP/G50029/1. NMO is funded by the Herchel Smith Fellowship. AGF is funded by the Vice Chancellor's Fellowship from the University of Sheffield. KRF is funded by European Research Council Grant 787932 and Wellcome Trust Investigator award 209397/Z/17/Z. This project is supported by a Templeton World Charity Foundation grant.

## Additional information

### Funding

| Funder | Grant reference number | Author |
|---|---|---|
| EPSRC | EP/G50029/1 | Rene Niehus |
| European Research Council | 787932 | Kevin R Foster |
| Wellcome Trust | 209397/Z/17/Z | Kevin R Foster |
| Wellcome Trust | Interdisciplinary Fellowship | Nuno M Oliveira |
| Biotechnology and Biological Sciences Research Council | BB/T009098/1 | Nuno M Oliveira |
| PKU-Baidu | 2020BD017 | Aming Li |
| College of Engineering, Peking University | Start-up funding | Aming Li |

The funders had no role in study design, data collection and interpretation, or the decision to submit the work for publication.

### Author contributions

Rene Niehus, Conceptualization, Resources, Data curation, Software, Formal analysis, Validation, Investigation, Visualization, Methodology, Writing - original draft, Writing - review and editing; Nuno M Oliveira, Conceptualization, Formal analysis, Investigation, Methodology, Writing - review and editing; Aming Li, Validation, Methodology, Writing - review and editing; Alexander G Fletcher, Formal analysis, Supervision, Investigation, Writing - review and editing; Kevin R Foster, Conceptualization, Investigation, Methodology, Writing - original draft, Project administration, Writing - review and editing

### Author ORCIDs

Rene Niehus (ID) https://orcid.org/0000-0002-6751-4124
Alexander G Fletcher (ID) http://orcid.org/0000-0003-0525-4336
Kevin R Foster (ID) https://orcid.org/0000-0003-4687-6633

### Decision letter and Author response

Decision letter https://doi.org/10.7554/eLife.69756.sa1
Author response https://doi.org/10.7554/eLife.69756.sa2

## Additional files

### Supplementary files

• Transparent reporting form

### Data availability

The MATLAB code for the regulated toxin model, the invasion analysis, and the evolutionary tournament is available on github (https://github.com/reneniehus/bact_warfare, copy archived at https://archive.softwareheritage.org/swh:1:rev:923e104aa634230547ba464c6bc8fee07f662ffa).

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

# Appendix 1

Appendix 1—code 1: Pseudo code to find the evolutionarily stable strategy of fixed toxin production.

```
Initialization: Find the initial resident fresini (for example, start with the highest average resident biomass)

forward = 1 indicates if the next migrant will have a higher or lower toxin investment than the resident. Can be 1 or -1.

previousforward = 1 record the direction of strategy change from the previous iteration

previousfres = 0 record the resident from the previous iteration

step = 0.1 initial step for the strategy change (initially coarse to localize the ESS)

minstep = 0.0001 the precision we want for the strategy value

singular = 0 boolean saying if a singular strategy is localised

nbflip = 0 record the number of consecutive flips in direction

fres = fresini

newres = 0 boolean saying if there is a new resident

while (!singular) {do this while no singular strategy is localised

  if (newres) {
    res vs N res competition. Compute the resident against the resident. Record the resident average strategy: wresav.
  }

  fmut = fres + forward*step pick an invader that differs from resident according to direction

  mut vs N res competition. Compute mutant vs resident competition. Record the mutant biomass (wmut) at the end of the competition.

  previousforward = forward update the previous direction
  previousfres = fres update the previous resident before the competition

  if (wmut < wresav) {resident stays the resident.
    forward = - forward change the direction, to test a migrant with lower f value at the next step
  }

  if (wmut > wresav) {the mutant invades and replaces the resident
    fres = fmut the new resident will take the migrant value
    newres = 1 we have a new resident, we'll have to compute its resident fitness at the next iteration
  }

  if (forward != previousforward) {compare the direction with previous direction
    nbflip = nbflip + 1
  }

  if (forward == previousforward) {compare the direction with previous direction
    nbflip = 0
  }

  this will allow to localise an ESS. We want fres to stay identical for 2 consecutive time steps, but we want to make sure that the
higher and lower strategies have been checked. Indeed if the mutant that we just tested goes extinct, the resident will stay resident
but that is not enough to say it is an ESS. We also want the number of flips in direction to be > 1.

  if (previousfres == fres & nbflip > 1) {singular strategy localised

    if (step <= minstep) {precision is high enough
      singular = 1 we have found the optimal strategy
    }
```

```
    if (step > minstep) {will redo the whole procedure with tinier step

        step = step/10 increase precision 10 times

        nbflip = 0 reset the number of consecutive flips

    }

  }

}

Record the final fres (ESS).

Finally, we test whether this fres (ESS) is a GLOBAL optimum by competing it against the range of strategies given through f = [0,

0.01, 0.02, ....1].
```

## Supplementary analytics

For the system of ODEs presented in *Equation (1)*, the system state could be rewritten in the vectorised form $\mathbf{X} = (C_A(t), C_B(t), T_A(t), T_B(t), N(t))^{\mathrm{T}}$ with the system dynamics denoted as

$$\frac{d\mathbf{X}}{dt} = \mathbf{F}(X, f_A, f_B, k, l_r). \tag{S1}$$

We denote a specific equilibrium state by $\mathbf{X}^*$. In order to analyse the associated linearly asymptotical stability of the above system at $\mathbf{X}^*$, we should first find solutions safisfying $\mathbf{F}(\mathbf{X}^*, f_A^*, f_B^*, k, l_T) = 0$. We note, that in *Equation (1)*, the dynamics of $N$ is defined by a negative derivative, and from this derivative we can see that a stable state regarding $N$ can only be reached when there are no cells or no nutrients. However, no cells is a trivial and extreme state (i.e. no cells are left for further seeding), and no nutrients cannot be reached within finite durations. We will therefore abandon the dyanmics of $N$ from the system, basically assuming chemostat environment where different levels of $N$ can be acchieved through balancing consumption and influx of $N$. This changes of course how cell strains interact, but it will help to show in a simpler way how analytical methods fail even for this simplified system of equations.

Now, setting $\mathbf{F}(\mathbf{X}^*, f_A^*, f_B^*, k, l_T) = 0$ further gives

$$(1 - f_A)u_{max}\frac{N}{N + k_N}C_A - kT_BC_A = 0 \tag{S2}$$

$$(1 - f_B)u_{max}\frac{N}{N + k_N}C_B - kT_AC_B = 0 \tag{S3}$$

$$f_A\frac{N}{N + k_N}C_A - l_TT_A = 0 \tag{S4}$$

$$f_B\frac{N}{N + k_N}C_B - l_TT_B = 0 \tag{S5}$$

From *Equation (S2)*, we have

$$T_B = (1 - f_A)u_{max}\frac{N}{k(N + k_N)}. \tag{S6}$$

From *Equation (S3)*, we have

$$T_A = (1 - f_B)u_{max}\frac{N}{k(N + k_N)}. \tag{S7}$$

From *Equation (S5)*, we have

$$T_B = f_B\frac{N}{(N + k_N)l_T}C_B. \tag{S8}$$

From *Equation (S4)*, we have

$$T_A = f_A \frac{N}{(N+k_N)l_T} C_A. \tag{S9}$$

Thus, we know from *Equations (S8) and (S9)* that

$$\frac{T_A}{T_B} = \frac{f_A C_A}{f_B C_B}. \tag{S10}$$

Similarly from *Equations (S6) and (S7)*, we get

$$\frac{T_A}{T_B} = \frac{1-f_B}{1-f_A}. \tag{S11}$$

Therefore, from *Equations (S10) and (S11)*, we have

$$f_A C_A (1-f_A) = f_B C_B (1-f_B). \tag{S12}$$

From *Equations (S6) and (S8)*, we know that

$$f_B = (1-f_A)u_{max}\frac{l_T}{kC_B}. \tag{S13}$$

From *Equations (S12) and (S13)*, we know

$$f_A^* = \frac{l_T \mu_{max} k C_B - \mu_{max}^2 l_T^2}{k^2 C_A C_B - \mu_{max}^2 l_T^2}, \tag{S14}$$

and note that here when $f_A^* = 1$, we obtain $f_B^* = 0$, which is not applicable.

Substituting *Equation (S14)* into *Equation (S9)*, we then know

$$T_A^* = \frac{\mu_{max} k C_B - \mu_{max}^2 l_T^2}{k^2 C_A C_B - \mu_{max}^2 l_T^2} \frac{N}{N+k_N} C_A. \tag{S15}$$

Similarly, we further have

$$f_B^* = \frac{l_T \mu_{max} k C_A - \mu_{max}^2 l_T^2}{k^2 C_A C_B - \mu_{max}^2 l_T^2}, \tag{S16}$$

and

$$T_B^* = \frac{\mu_{max} k C_A - \mu_{max}^2 l_T^2}{k^2 C_A C_B - \mu_{max}^2 l_T^2} \frac{N}{N+k_N} C_B. \tag{S17}$$

And the Jacobian matrix of the system state $\mathbf{X}^*$ is

$$A = \frac{\partial \mathbf{F}}{\partial \mathbf{X}}|\mathbf{X}^* = $$

$$\begin{bmatrix} (1-f_A)\mu_{max}\frac{N}{N+k_N} - kT_B & 0 & 0 & -kC_A & \frac{k_N}{(N+K_N)^2}(1-f_A)\mu_{max}C_A \\ 0 & (1-f_B)\mu_{max}\frac{N}{N+k_N} - kT_A & -kC_B & 0 & \frac{k_N}{(N+K_N)^2}(1-f_B)\mu_{max}C_B \\ f_A\frac{N}{N+K_N} & 0 & -l_T & 0 & \frac{k_N}{(N+K_N)^2}f_A C_A \\ 0 & f_A\frac{N}{N+K_N} & 0 & -l_T & \frac{k_N}{(N+K_N)^2}f_B C_B \\ -\frac{N}{N+K_N} & -\frac{N}{N+K_N} & 0 & 0 & -\frac{N}{N+K_N}(C_A+C_B) \end{bmatrix} \tag{S18}$$

By computing the maximum real part of all the eigenvalues of $\mathbf{A}$ (denoted by $Re(\lambda)$), we know that $\mathbf{X}^*$ is linearly asymptotical stable when $Re(\lambda)<0$. This canonical measure of stability indicates whether and how fast the system would return to $\mathbf{X}^*$ after a small perturbation (*Hofbauer and Sigmund, 1998*; *Mougi and Kondoh, 2012*; *Coyte et al., 2015*; *Allesina and Tang, 2012*).

```
mu = 10;
k = 20;
l = 0.1;

Ca = 0.1;
Cb = 0.1;
fa = 0;
deltafa = 1;

fb = 0;
deltafb = 1;

Ta = 1;
Tb = 1;
N = 1;
```

```
Endtime = max_end_time; % given in hours
TIME = [0,Endtime];
dt = 0.1;        % time step for numerical solution
param.gamma = 1; % the consumption of nutrients

param.KN = 5; % half-saturation constant for nutrient-dependent growth
param.mu = 10; % max growth rate
param.kay = 20; % how many cells are killed per unit toxin
param.D = 0.10; % % diffusion loss of toxin in ODE
param.Ca0 = 0.1; param.Cb0 = param.Ca0;

Ta0 = 1; Tb0 = 1;  % define initial conditions
param.N0 = 1;
```

different values of $C_A^*(t)$ and $C_B^*(t)$ ranging from 0.1 to 1. We choose given in **Table 1**, and $f_A^*$, $f_B^*$, $T_A^*$, and $T_B^*$ take values defined in

We could see that all of possible values of $C_A$ and $C_B$ indicate that the system is unstable, suggesting that an analytical analysis to exactly capture the temporal unstable state of the system is not applicable.

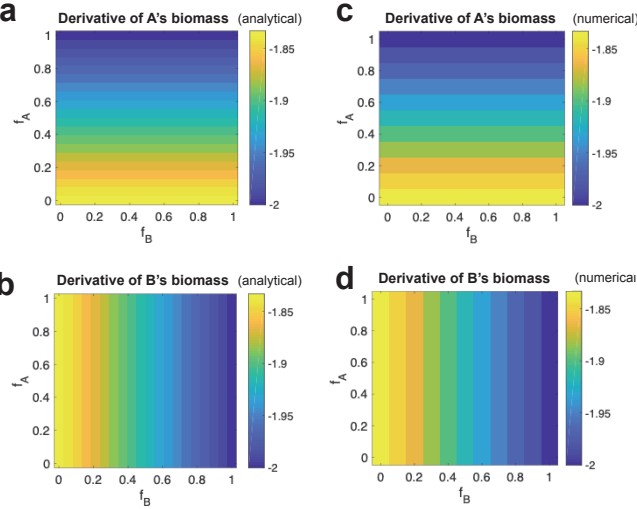

**Appendix 1—figure 1.** Analytical derivative of strain biomass compared to numerical change in strain biomass. For different combinations of $f_A$ and $f_B$ values, analytical derivative of strain $A$'s biomass (**a**) and of strain $B$'s (**b**). This is shown side-by-side with numerical change in biomass of strain $A$ (**c**) and strain $B$ (**d**).

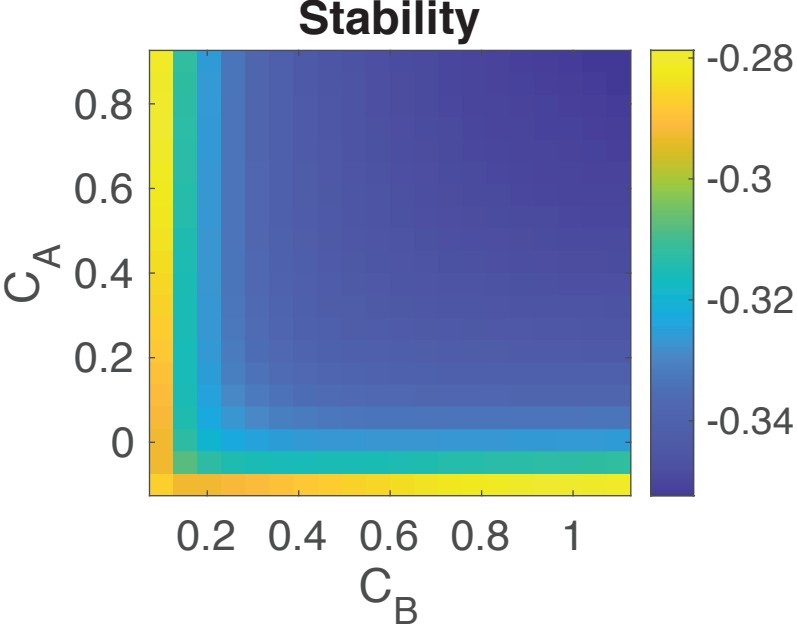

**Appendix 1—figure 2.** Stability analysis of the system. Plot shows for a range of values for $C_A$ and $C_B$ the negative of the maximum real part of all the eigenvalues of $\mathbf{A}(-Re(\lambda)$, see details above). Negative values indicate that the system's state is stable. We choose values for $k$ and $l_T$ to be as given in *Table 1*.

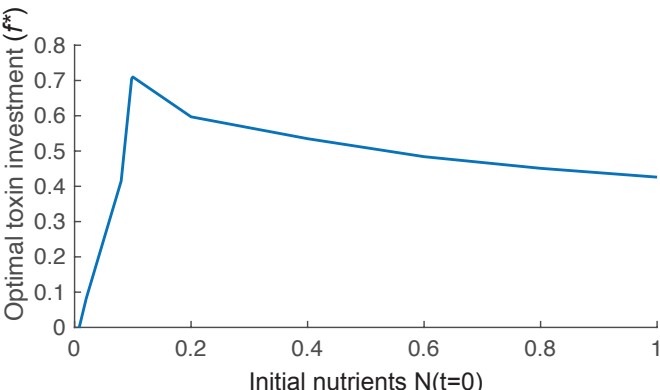

**Appendix 1—figure 3.** The effect of nutrient availability on optimal toxin investment. We plot the evolutionary stable investment into toxin over a range of different initial levels of nutrient ($N(t=0)$). The optimal investment is highest for an intermediate amount of nutrients. Other parameters of the model take the standard values given in *Table 1*.

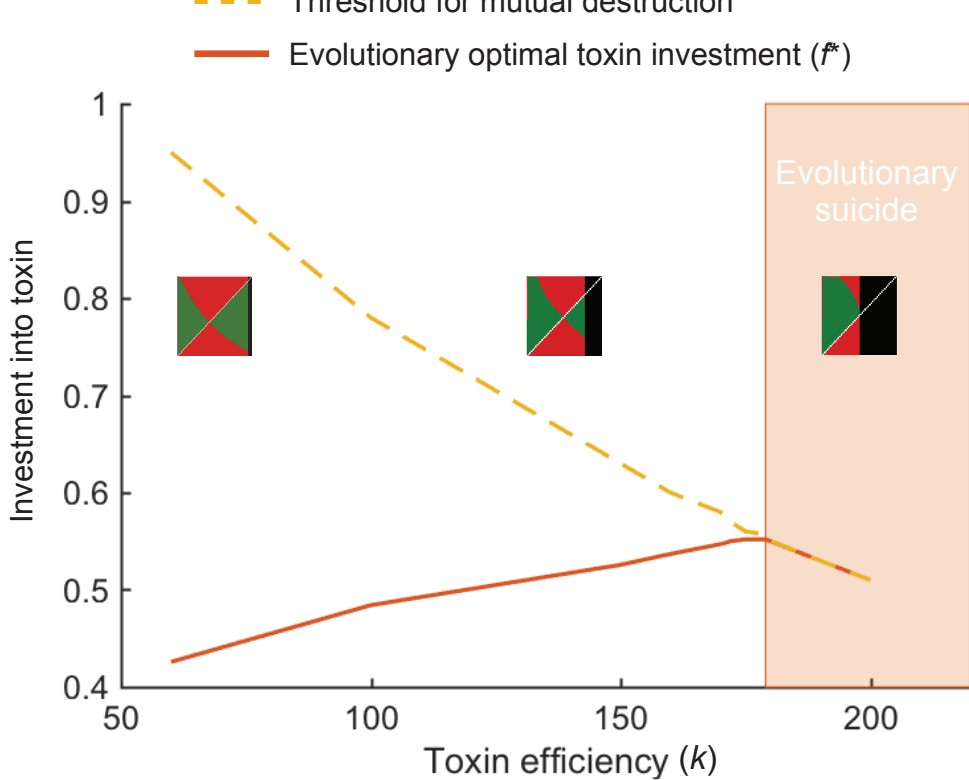

**Appendix 1—figure 4.** At high toxin efficiency an evolutionary arms race can drive populations towards extinction. We plot the optimal toxin investment $f^*$ (red line) and the investment threshold for mutual destruction $f_{kill}$ (orange line) over different toxin efficiencies (**k**). Insets show the pairwise invasibility plot for low, medium, and high toxin efficiency. In these plots the x and y axis give the toxin investment (with range 0–1) of the resident and the invading strategy, respectively. Green areas indicate where the invader strategy can successfully replace the resident strategy, red areas indicate where invasion fails. Where the two green and two red triangles have their meeting point at the diagonal line, lies the evolutionarily stable strategy. Black areas indicate where the resident strategy dies off in competition with itself. We see that above a certain toxin efficiency ($k = 179$), coevolution causes toxin strategies to produce amounts of toxins that are deadly to both competitors. The toxin arms race causes an evolutionary suicide of the population, analogous to mutually assured destruction. All other model parameters take standard value as given in *Table 1*.

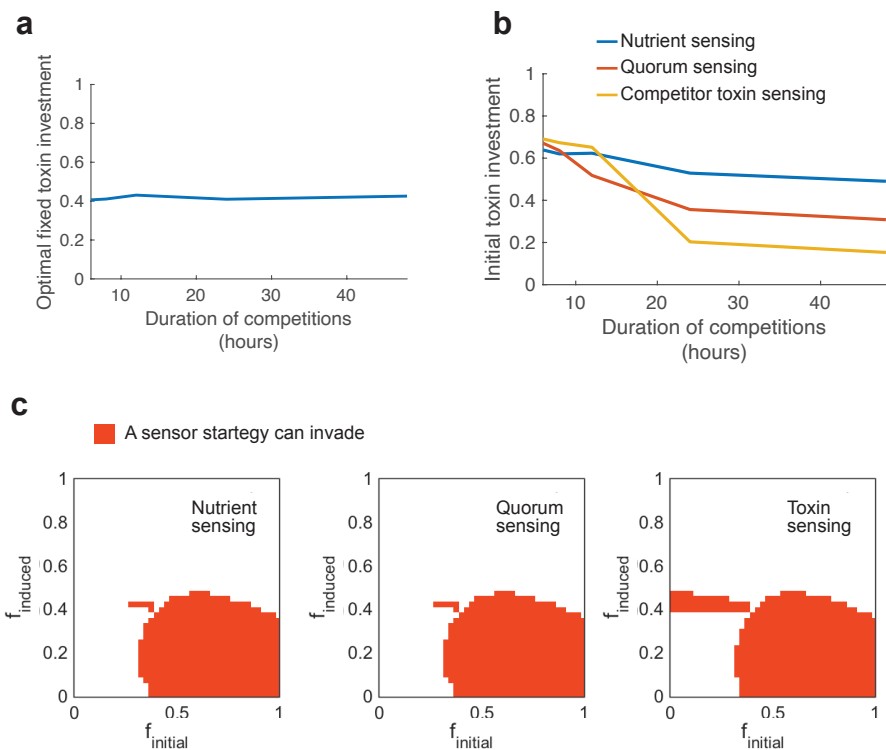

**Appendix 1—figure 5.** Short competitions favour the evolution of pre-emptive attack. We investigate the effect of shortening the duration of competitions and ask how this affects the best performing strategies. In nature, the duration of competition will vary depending on the rates of dispersal to new patches. (**a**) Shortening competition time has little effect on the evolution of constitutive toxin production. (**b**) Initial investment in regulated toxins increases strongly, favouring pre-emptive attacks. Short competition times select for an increased baseline of aggression in sensing strategies, because it becomes important to overcome a competitor as quickly as possible. (**c**) At short competition times (6 hr), regulation still remains beneficial and strategies of all three sensing types exist that can invade and evolutionarily replace all constitutive producers (red areas). All parameters take standard values as given in *Table 1*.

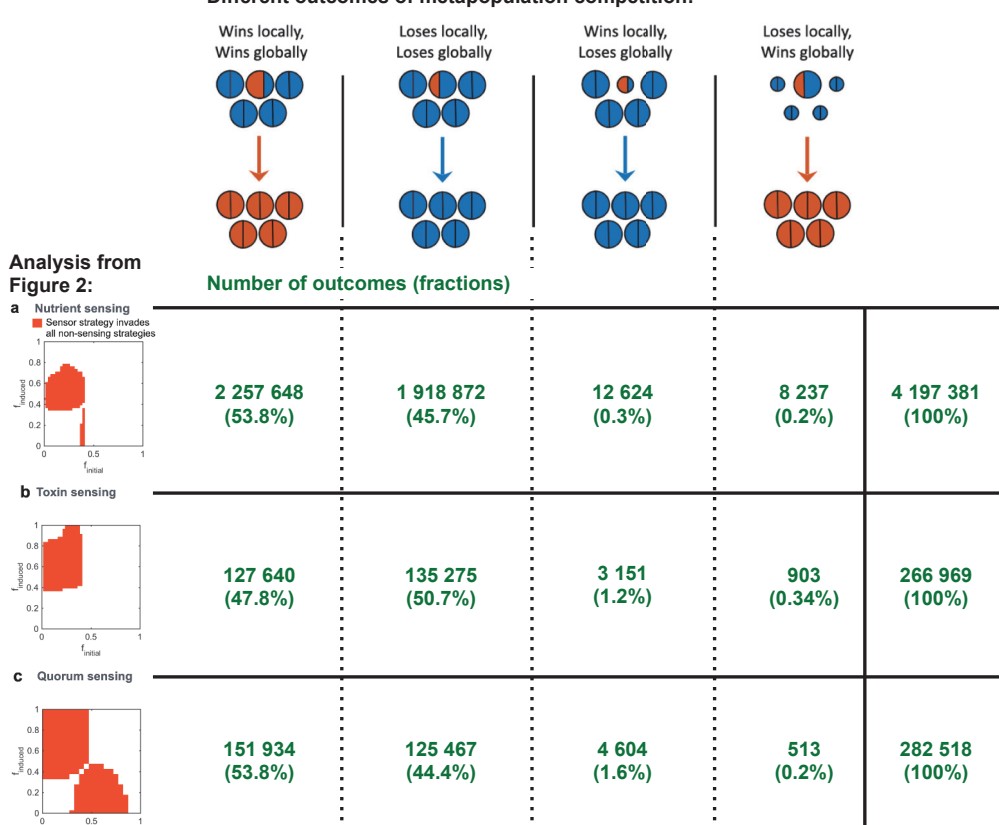

**Appendix 1—figure 6.** All four competitive dyanmics occur in our simulations. We enumerate the occurrence of the four different outcomes from the global and local competitions as shown in *Figure 1d*. Here, the numbers (shown in green) of these outcomes are shown for the simulations used for the three different analyses in *Figure 2*. We also show in the column to the right the total number of simulations run for each of these analyses, and we give in brackets the proportion of occurrences relative to this total number.

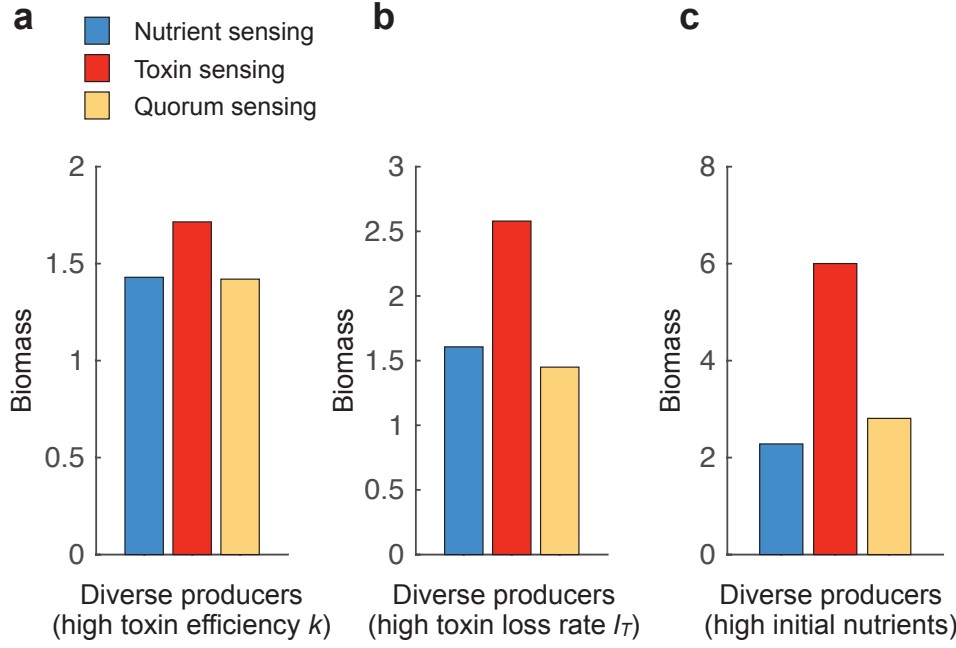

**Appendix 1—figure 7.** The benefit of toxin sensing is robust across varying environmental conditions. *Figure 3* shows that toxin sensing is the best performing strategy when competing with diverse toxin strategies. Here, we compete the different sensing strategies again against a range of nine constitutive producers ($f$=0.1, 0.2, . . ., 0.9) (as in *Figure 3b*) but under different conditions, which are (**a**) high killing efficiency of the toxin ($k$=30), (**b**) high loss rate of the toxin ($l$=0.4), and (**c**) high initial nutrients ($N(t$=0)=5). All other parameters take standard values as in *Table 1*.

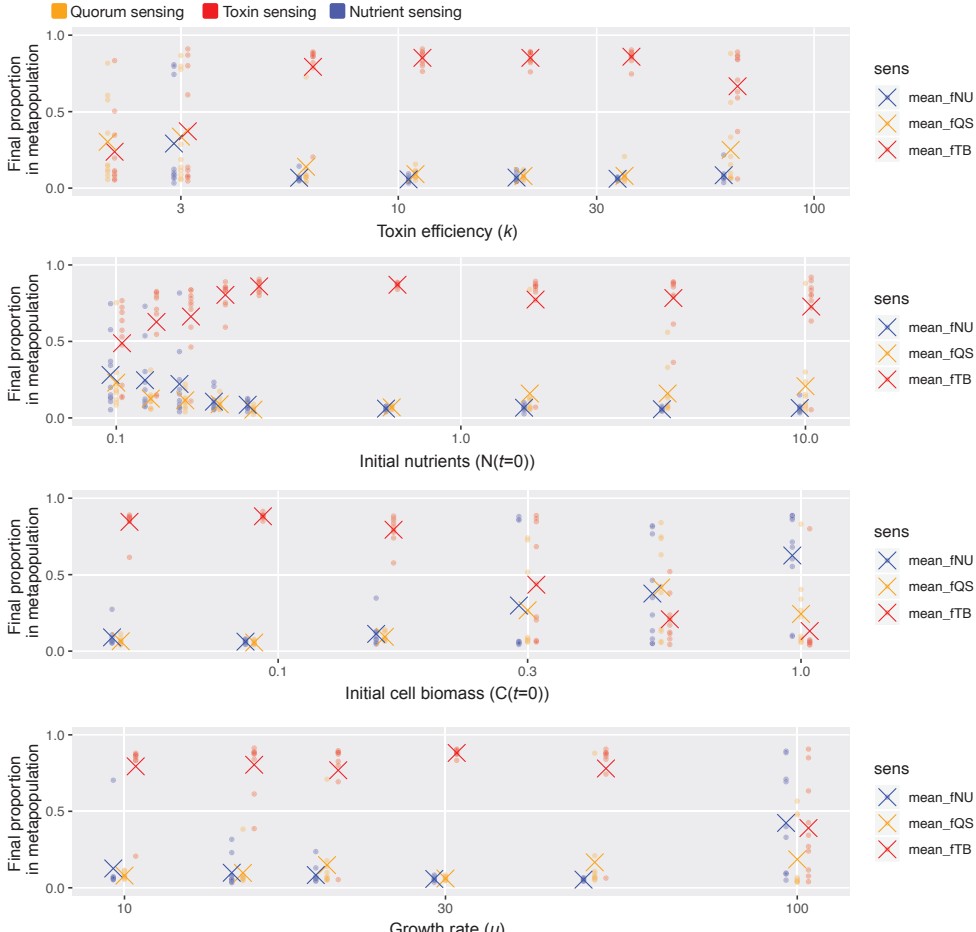

**Appendix 1—figure 8.** Toxin sensing emerges as the overall winner across wide parameter ranges. As in *Figure 4c*, we pit all three sensory strategies against each other in a coevolutionary tournament (genetic algorithm) and we record the proportion of each of the three different types of strategies at the end of the evolution. We show these proportions as coloured dots, and the average proportion across 10 repeat runs of the algorithm as coloured crosses. Finally, we repeat this while varying one model parameter at a time over approximately one order of magnitude, keeping the other parameters as given in *Table 1*. Toxin-based regulation only fails to show dominance under parameter regimes where selection for the different strategies is relatively weak and noisy in its outcome, that is, low *k* (toxin has weak effect), high initial cell numbers/low nutrients/ slow cell division (few cell divisions per competition cycle and so weaker natural selection).

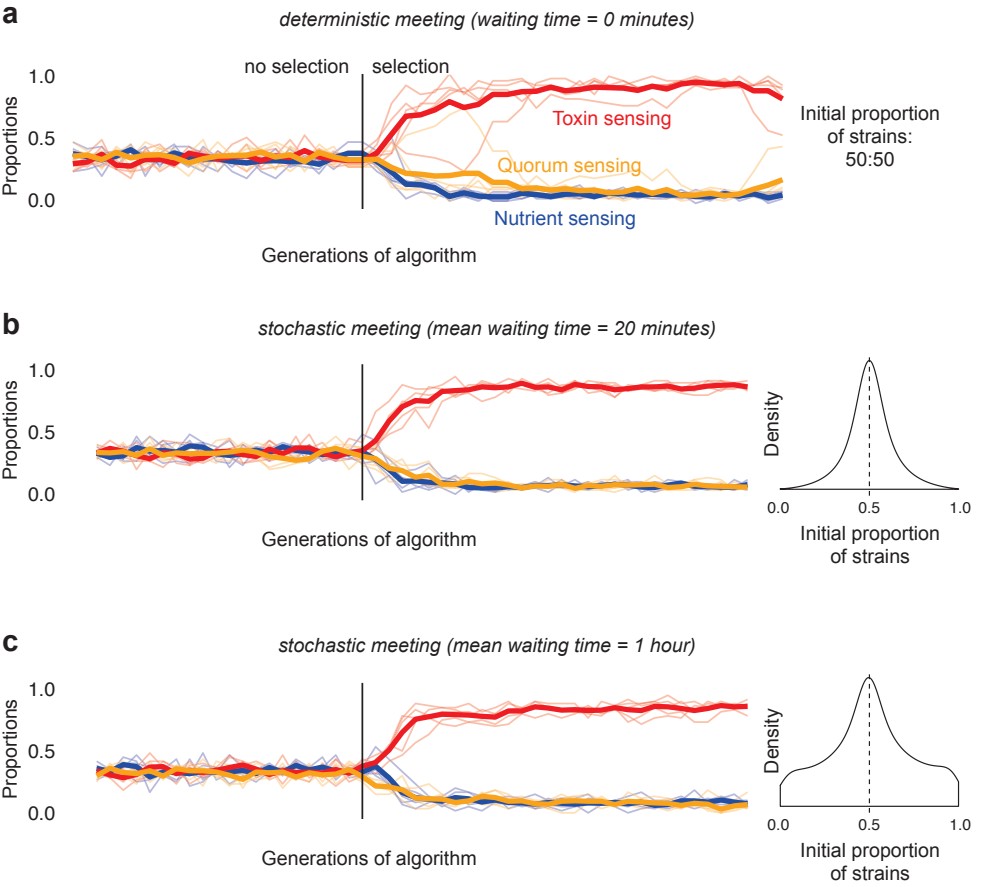

**Appendix 1—figure 9.** The evolution of reciprocation (toxin responders) in models of deterministic and stochastic initial strain frequency. (**a**) Outcome of the large tournament model with initial strain proportion at 50:50 (as in *Figure 4*). Shown is the metapopulation proportions of the three different strategy types (toxin sensing in red, quorum sensing in orange, nutrient sensing in blue) over time. (**b**) and (**c**) show the same tournament but with competing genotypes arriving stochastically into each competition to create a wide range of initial proportions of each strain ranging from 0 to 1. Density plots on the right show the distribution of initial proportion of strains. The arrival order of the two competing strains is chosen uniformly at random, then the waiting time until the next strain arrives follows an exponential distribution with mean of 20 min (**b**) and 1 hr (**c**). We see that, both for the 50:50 mix (**a**), and under variable initial frequency (**b and c**) the toxin responders evolve (red line), rather than quorum- (yellow) or nutrient-sensing (blue) strains.

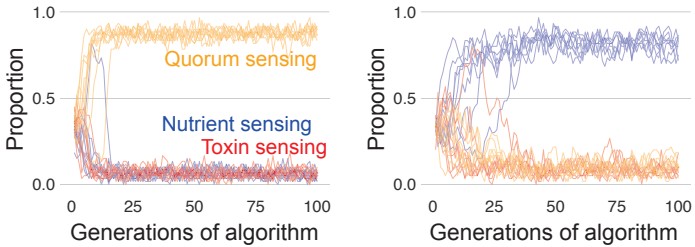

**Appendix 1—figure 10.** The benefit of toxin sensing over other sensing strategies is lost when competition time is short. As in *Figure 4c*, we pit all three sensory strategies against each other in a
*Appendix 1—figure 10 continued on next page*

*Appendix 1—figure 10 continued*

coevolutionary tournament, but with shortened competition time of 6 hr, which is before the toxin sensor typically recovers (*Figure 4d*). Lines show the population proportions of the three strategies for 10 runs of the tournament. In the left-hand side, panel quorum sensing is most successful, but which strategy wins changes based upon chance variation in the initial parameter values such that nutrient sensing (right-hand side panel) and toxin sensing (not shown) can also win depending on the parameter ranges set for each sensing type. This indicates that with short competition time, who wins is determined by lucky initial parameter draws that allow one type to dominate the population with a pool of optimised individuals, but the competitive optimum can be reached equally by any of the three sensing types.

