## [Decision Letter]

**Acceptance summary:**

Bacteria often produce toxins to fight competitors. There has been strong interest in understanding the molecular mechanisms of toxin production, release and mode of actions. Less well understood are the costs and benefits of toxin production and how natural selection acts on regulatory circuits controlling toxin production. The paper tackles this problem. Using computer simulations, the authors show that regulated toxin production is generally a better strategy than constitutive toxin production, and reciprocication is fundamental in competitions. Interestingly, reciprocication doesn't evolve into peaceful coexistence, but a strenuous battle.

**Decision letter after peer review:**

[Editors’ note: the authors submitted for reconsideration following the decision after peer review. What follows is the decision letter after the first round of review.]

Thank you for submitting your article "The Evolution of Strategy in Bacterial Warfare" for consideration by *eLife*. Your article has been reviewed by 3 peer reviewers, and the evaluation has been overseen by a Reviewing Editor and a Senior Editor. The following individual involved in review of your submission has agreed to reveal their identity: Rolf Kümmerli (Reviewer #3).

Our decision has been reached after consultation between the reviewers. Based on these discussions, we regret to inform you that we cannot accept your work for publication in *eLife* at present. However, if you address all the major concerns below through in-depth changes to the manuscript and new computations, we will be happy to consider a suitably revised manuscript as a new submission, that will be sent back to the same referees. Suggestions below were synthesized from the 3 reviews by the Reviewing Editor.

Summary:

Bacteria typically produce toxins to fight competitors. There has been strong interest in understanding the molecular mechanisms of toxin production, release and mode of actions. Less well understood are the costs and benefits of toxin production and how natural selection should act on regulatory circuits controlling toxin production. The paper tackles this problem. Using computer simulations, the authors show that regulated toxin production is generally a better strategy than constitutive toxin production, and reciprocication is fundamental in competitions. Interestingly, reciprocication doesn't evolve into peaceful coexistence, but a strenuous battle.

All Reviewers and the Reviewing Editor found that the paper makes important and interesting conclusions, and agreed that the model is relevant. However, there were significant reservations about methodology, and about whether the approach fully warrants the conclusions of the paper.

Essential revisions:

1) Because of the computational approach of searching parameter space of a quite complex ODE model, a clear quantitative understanding is not provided, but rather an intuition for the observed results. While the model captures the different toxin regulatory systems, which may be complicated to tackle analytically, some analytical insight would really help to demonstrate the generality of the conclusions.

For instance, an analytical model contrasting constitutive vs. regulated traits (first part of the paper), would build a stronger foundation.

2) The authors should clarify why the local versus global analysis is required. This is all the more true that the effect of spatial structure was not explored in the current paper.

If all competitions are pair-wise and the main focus are strategies that invade all other strategies and cannot be reinvaded, why is a metapopulation analysis necessary? Also, the concept is presented is presented in Figure 1, but it is not farther discussed.

The strategies that win/lose locally, but lose/win globally should be discussed to shed light on the importance of this metapopulation analysis. This might matter with regards to possible extensions of the model to spatial settings.

3) The connection to (evolutionary) game theory appears superficial. The authors should clarify this point and be particularly careful about wording when they explain the bases of their model.

In particular, the sentence "we combine evolutionary game theory with differential equation modelling" (line 20) is unfortunate. Evolutionary game theory is primarily differential equation modelling, as for instance, the replicator and replicator mutator equations are ordinary differential equations. (One exception regards finite population analysis, where tools from statistical physics enter.)

In addition, invasion analysis by itself is static game theory. It implies dynamics from stability of fixed points, but not the actual dynamics itself which is taken into account when studying evolutionary games.

4) Are the initial frequencies of the two competing populations important for the final outcome? This is related to the previous question regarding different outcomes between local and global. Also, while the model is currently deterministic, in real case scenarios noise plays a big role, so it would be good to briefly discuss this.

5) In the regulated models, the parameter constraints allow f_induced and f_initial to range between 0 and 1. But because of how the model is set-up, this means that it can happen (and it definitely does looking at Figure 2) that f_initial+f_induced=f is outside the [0,1] range, which is though the constrained range for the f of the constitutive competing strain. This might cause strange model behavior and definitely an unfair competition, so these parameters should be removed/checked.

1) The model developed in the manuscript captures the toxin regulatory system in bacteria, which is very interesting. However, the title of the manuscript and the abstract should be revised to better reflect the specific system under study.

2) The population dynamic equations in eq set 1 could be analysed further to get some analytical handle to promote our understanding. The authors could start from the following paper Vasconcelos, P., Rueffler C., 2020, How Does Joint Evolution of Consumer Traits Affect Resource Specialization? The American Naturalist 195: 331-348.

In addition, the final biomass densities used in the calculation of the invasion fitness (Eq. 2) could be further simplified by focusing on the equilibrium solutions of Eqs. 1 (perhaps under which assumptions the solutions are possible could already be interesting). Providing an expression for the invasion fitness would be a real plus.

3) Please clarify the rationale for choosing parameters in the simulations, and discuss robustness, beyond the statement "parameters of the algorithm […] are chosen to achieve short simulation times and good convergence behaviour as determined by visually inspecting the distribution of population parameters over time." (727-730)

4) Some of the findings are intuitive, as the authors acknowledge. E.g. regulated traits outperform non-regulated traits, but others are more surprising and very interesting. For example, many toxins are quorum-sensing regulated (e.g. phenazines in *P. aeruginosa*). But the authors show that this is not an ideal mechanism because a strain might be killed before it's reaching a high enough density to switch on QS-regulated toxins. I'd like to see a more detailed discussion on this putative mismatch. Related to this, my intuition is that the three mechanisms might work in concert, i.e. a toxin is QS controlled, but the threshold for QS activation is lowered by competition sensing. This would be interesting to discuss.

5) The conclusion on lines 32-35 is strong. It might apply to mixed conditions as simulated here. However, in spatially structured habitats reciprocal fighting might lead to co-existence as competitors manage to defend their 'territories' and fighting only occurs at the borders. The swift elimination of competitors is maybe less common than assumed in real-world set ups.

6) The argument is that toxin sensing works best as the winner switches off toxin production once the competitor is eliminated. However, toxins might outlast their producers, such that the switching off might take longer than assumed. Can the authors comments on this?

[Editors’ note: further revisions were suggested prior to acceptance, as described below.]

Thank you for submitting your revised article "The evolution of strategy in bacterial warfare: quorum sensing, stress responses, and the regulation of bacteriocins and antibiotics" for consideration by *eLife*. Your article has been reviewed by 2 peer reviewers, and the evaluation has been overseen by a Reviewing Editor and Aleksandra Walczak as the Senior Editor. The following individual involved in review of your submission has agreed to reveal their identity: Rolf Kümmerli (Reviewer #2).

Both reviewers are very positive about your revised manuscript and recommend publication. However, both reviewers and the reviewing editor agree that one revision would improve your manuscript. Therefore, we would like to ask you to do this revision before we accept your manuscript for publication.

Essential revision:

While the motivation for the local vs. global analysis is well explained, the results from this analysis should be clarified. In figure 1, interesting examples are proposed of strategies that might win/lose locally and then lose/win globally. These are very interesting cases that point at non-trivial competition dynamics. It would be important to add a figure or table (perhaps in the supplement) and an associated discussion paragraph where it is shown that doing this additional meta-population analysis is necessary and adds something to the local competition part. The aim of this addition would be to address the following question: Would the optimal strategy change if one didn't do the global competition step? Some statistics of counter-intuituive scenarios explored, based on their classification in figure 1d, would address this point.

*Reviewer #1 (Recommendations for the authors):*

The manuscript puts forward exciting hypotheses about the strategies of toxin production in bacteria, which would be very interesting to test experimentally. Also, future work that tries to shed deeper analytical insight in the results would undoubtedly be very interesting.

I think the work is definitely worth publication. I find the presentation of the results sometimes difficult to follow, partially because the details of the models, which is very helpful to understand the results, is detailed at the end of the paper in the methods. But probably this is unavoidable given the format.

I believe the current version of the manuscript reads better and clarifies some of the misunderstandings in previous versions. There are, however, some criticisms that have not, in my opinion, been well addressed.

1) I appreciate the complexity of the model and the strengths that come with including this complexity. The new analytical work carried out to investigate stability of the fixed points helps towards the analytical interpretation of the results. I think, however, the criticism previously raised by the reviewers was trying to determine whether a simpler model, analytically tractable, would be able to reproduce some of the results showed here, while giving more analytical insight. I don't think the manuscript currently addresses this issue. On the other hand, I also think that it can be left to future work as it is a very interesting and challenging research direction.

2) I am satisfied with the motivation behind the local versus global analysis, which I think is very important, especially in potential future applications to spatial settings of this work. The motivation behind the analysis was never in question. What I don't understand are the results from this analysis. In figure 1, interesting examples are proposed of strategies that might win/lose locally and then lose/win globally. These are very interesting cases that point at non-trivial competition dynamics that would be interesting to investigate further in future work, but I don't see these cases discussed anywhere in the results. I understand that the results of the algorithm come from a sequence of local competition, dispersal, seeding, etc…, that include this metapopulation competition, but I would like to see a figure/paragraph/discussion where it is shown that doing this additional meta-population analysis adds something to the local competition part. Would the optimal strategy change if one didn't do the global competition step? If the answer is yes, which I imagine it is, where do I see this?

*Reviewer #2 (Recommendations for the authors):*

The authors have done a very good job in revising their paper. The main issue that arose during the first round of reviewing was the lack of an analytical model to establish a stronger foundation of the principles of competition sensing. Although it was not me who brought up this issue, I have consulted the authors' responses, edits in the main paper and extra analysis in the supplements with great care. My opinion is that the authors have convincingly solved the debate. There is simply no possibility to device an analytical model that captures even the simplest regulatory circuits involved in competition sensing and toxin production. The most important thing is that the authors have not only used verbal arguments to make their point, but have immensely invested in analytical modelling to show why the approach does not work. I believe that the strength of the paper is its biological realism and the fact that it generates predictions that can be empirically tested.

Moreover, the authors have adequately addressed my own comments. The discussion on interactions between regulatory mechanisms and the role of ecology have significantly improved the paper. The addition on local vs. global interactions is also important, especially for non-specialist readers.

---

## [Author Response]

[Editors’ note: the authors resubmitted a revised version of the paper for consideration. What follows is the authors’ response to the first round of review.]

Essential revisions:1) Because of the computational approach of searching parameter space of a quite complex ODE model, a clear quantitative understanding is not provided, but rather an intuition for the observed results. While the model captures the different toxin regulatory systems, which may be complicated to tackle analytically, some analytical insight would really help to demonstrate the generality of the conclusions.For instance, an analytical model contrasting constitutive vs. regulated traits (first part of the paper), would build a stronger foundation.

We agree that analytical solutions are desirable whenever possible. We did not go this route in this study because our goal is to capture the details of bacterial regulation and competition that microbiologists care about, and to study when and why such complex regulation evolves. The issue then is that analytical solutions would require much simpler starting equations, or simplifying assumptions, with which we would not have achieved our goal. This is already evident in our equation 1, which is based upon a previously published study on bacterial competition via toxins (Bucci, Nadell, and Xavier 2011). This equation, although it does not include the details of toxin regulation, is already of a class that is problematic for analytics.

Nevertheless, over these last months, we have engaged in a serious effort to see what explorations might be possible analytically with our model or a simplified version of it that still captured the relevant biology. To do this, we brought on board a mathematician (Aming Li) who is an expert in the analytics of ecological and evolutionary processes. More specifically, we used Equation 1 from our manuscript and wrote the analytical form that gives the system's derivative for a given state with respect to time as well as the Jacobian matrix to explore the stability of the system (new Supplementary analytics section of the manuscript). It is important to note that we are making the simplifying assumption of setting nutrients to a constant, akin to a chemostat version of our model. We cannot make a similar assumption regarding toxins, as this would remove the key way in which the bacterial strains are interacting.

We then analytically explored the system's stability for a set initial conditions and a range of toxin investments. We compared the analytical results for the derivatives with the results from our numerical simulation, and we found them to be identical (Supplementary analytics section, Supplementary Analytics Figure 1). This is then a confirmation that analytics and our numerics give the same results, although this result is expected because we have implemented our numerical simulations with step sizes chosen to extremely accurately approximate an analytical solution. Our stability analysis of the system shows that all equilibria are unstable (Supplementary analytics section, Supplementary Analytics Figure 1), which is because our model contains feedback loops that render it unstable for non-trivial states.

Supplementary Analytics Figure 1 also demonstrates clearly how analytical explorations fall short of tackling the essence of our system: the results show that the toxin investment of strain A at time t does not impact biomass B, which seems counterintuitive. This is because the analytical solution explores an instant in time, but fails entirely to capture the key interactions between strains. Those occur through toxin production (and in the full model through consumption of limited nutrients), and those create feed-back loops within the system (for example toxin investment of A is responsible for reduction in biomass of B and then in turn for a reduction in toxin B), analogous to more classical predator-prey models, which we note have previously been shown to be intractable by analytics (Boys, Wilkinson, and Kirkwood 2008; Liu and Chen 2003; Gutiérrez and Rosales 1998).

In summary, we found that an analytical exploration of the model did not capture the key processes at hand, nor did it help us with our understanding. We have now added a new paragraph where we discuss this ("Overview" section of "Results and Discussion") and better explain why we rely on numerics. We also now include the analytical stability analysis as a Supplementary Method (Supplementary Analytics section).

In place of analytical results, we have numerically explored the evolution of different regulatory strategies under a wide range of conditions, including i) against one optimised unregulated strategy (Figure 2), ii) against a diversity of unregulated strategies (Figure 3), iii) against the same class of unregulated strategy (Figure 4a,b) and, finally, iv) against an open set of all of the regulated and unregulated strategies (Figure 4c,d). Across all of these scenarios, the strategy class that performs the best is toxin sensing, our key conclusion. Moreover, we have tested the robustness of this prediction across numerous parameter sweeps (Figures 2, S4, S5, S6), which again predict the supremacy of toxin sensing. Finally, we are able to provide an intuitive post hoc explanation for why toxin sensing does so well – it is able to downregulate toxin production after winning a competition – and, importantly, we demonstrate this explanation is indeed causal by shortening competition time which removes this advantage (Figures 3, S7). All of these points combine to give us a lot of confidence in our predictions.

To close this discussion, we would also like to emphasise that – while our work does not have proofs of the sort that are seen in some papers in theoretical biology – our attention to biological details does mean that we generate readily testable prediction e.g. bacteria should often use damage or cues of damage to coordinate their attacks. Our approach then allows us to fulfil our goals better than generating simpler models that lack these details in order to look for more abstract principles. That is, while our work may not be amenable to robustness testing via analytics, it *is* amenable to experimental testing by microbiologists, and this is our research priority.

2) The authors should clarify why the local versus global analysis is required. This is all the more true that the effect of spatial structure was not explored in the current paper.If all competitions are pair-wise and the main focus are strategies that invade all other strategies and cannot be reinvaded, why is a metapopulation analysis necessary? Also, the concept is presented is presented in Figure 1, but it is not farther discussed.The strategies that win/lose locally, but lose/win globally should be discussed to shed light on the importance of this metapopulation analysis. This might matter with regards to possible extensions of the model to spatial settings.

Our modelling captures different strategies for bacterial interactions and we are interested in the expected long-term evolutionary fate for these strategies. To achieve this in a way that is relevant to the real world, we consider the potential competition both within and between patches of bacteria, which recognises that the outcome of local competition in any given patch will often not be sufficient to predict evolutionary trajectories. Consider, for example, a competition between two strains of bacteria in one of the pores of your tongue. If we focus solely on local competition within the pore, then any strategy that results in a focal strain making more cells than its competitor will be favoured, even if this leads to relative ruin for the winning strain, with only a few cells surviving the process. However, given these competitions can go on in many pores across many tongues, it is unlikely that such extreme strategies would be favoured as they mean few cells are produced to colonise new patches. Instead, the best strategies are those that make the most cells to disperse, which can also mean they win locally, but it might not. This is what our global analysis captures and it is why it is critical for our modelling. To make this clearer, we now include the above thought experiment in the main text (under section "Evolution of warfare via unregulated toxin production") in reference to Figure 1. We also now discuss spatial structure both with regards to the local (ODE) model, and with regards to our metapopulation level (see "Conclusions", second paragraph).

3) The connection to (evolutionary) game theory appears superficial. The authors should clarify this point and be particularly careful about wording when they explain the bases of their model.In particular, the sentence "we combine evolutionary game theory with differential equation modelling" (line 20) is unfortunate. Evolutionary game theory is primarily differential equation modelling, as for instance, the replicator and replicator mutator equations are ordinary differential equations. (One exception regards finite population analysis, where tools from statistical physics enter.)In addition, invasion analysis by itself is static game theory. It implies dynamics from stability of fixed points, but not the actual dynamics itself which is taken into account when studying evolutionary games.

We agree that our wording in the example above and in other places was not very clear. However, our work is more than superficially linked to game theory. To appreciate this, one has to realise that there are, by now, two rather different approaches used in evolutionary game theory. Evolutionary game theory, as originally conceived by Maynard Smith and Price, is based on calculations that ask whether a given strategy will invade from initially being rare (static game theory e.g. Maynard-Smith 1982. Evolution and the Theory of Games, Cambridge University Press). These are the kinds of tests that we are performing. More recently, an alternative set of approaches that use differential equations to follow the evolutionary process have also been called evolutionary game theory (dynamic game theory e.g. Hofbauer and Sigmund 1998. Evolutionary games and population dynamics. Cambridge University Press).

The confusion here then lies in the fact that we are not using this second form of game theory, but we are using ODEs. However, in our case the ODEs follow competitions on ecological timescales, which we use to calculate the success of a strategy against another strategy who meet locally. It is because of these biological details – the competition for nutrients and with toxins and the resulting feedback loops – that we need to capture in our local competitions that our system is not amenable to using dynamic game theory (see our response to point (1)). And of course we cannot just stop at modelling local competition without studying how this will affect the entire meta-populations on an evolutionary time-scale (as discussed in the last point). To achieve this, we combine our numerical solutions of our local competition with static game theory to ask whether this local invasion will lead to a global invasion (Maynard Smith and Price). To avoid further confusion for any reader, we now make it clearer that we are using static game theory in the text as follows:

–line 33: Abstract “Here we combine a detailed dynamic model of bacterial competition with static game theory to study the rules of bacterial warfare.” This avoids too many detailed modelling terms before we define them clearly in the main text

(previously: “Here we combine evolutionary game theory with differential equation modelling to study the rules of bacterial warfare.”)

– line 95: Introduction “Here we study the evolution of strategy during bacterial warfare by combining a detailed differential equation model of toxin-based competition with static game theory to identify the most evolutionarily successful strategies.” (previously: “Here we study the evolution of strategy during bacterial warfare by combining an explicit differential equation model of toxin-based competition with evolutionary game theory.”)

– line 119: Results and Discussion, Overview “We use these differential equations to model ecological interactions of bacterial strains and determine the outcome of competition for a given strategy against another strategy when they meet locally. These local-level competitions are embedded into a larger meta-population framework that determines long-term evolutionary outcomes [56] (Figure 1c and d). This meta-population modelling includes invasion analysis, in the tradition of static game theory developed by Maynard Smith and Price [57], and a more explicit genetic algorithm that employs the same logic (Methods). This algorithm pits diverse strategies against each other across a large number of combinations in order to find the most successful strategies. In these meta-population models, bacterial strains are assumed to compete locally in a large number of patches, but also globally through dispersal to seed new, empty patches based on a standard life history of bacteria used in previous models [58–62] (Figure 1c). Also as previously [62], we refer to the global population as a metapopulation to distinguish it from the local bacterial cell population in each patch. This approach accounts for the possibility that a strategy can do well in local competition, but do poorly globally, and vice versa (Figure 1d). ” (previously: “We use these equations to pit different strategies of attack against one another, and these local competitions are imbedded into a larger framework that uses evolutionary game theory [55] to understand which strategies will evolve (Figure 1 c and d).”)

– line 225: “To capture this effect, we embed the local-level competitions within a broader meta-population framework in order to make evolutionary predictions [56,69,70] (see Figure 1 c and d). This framework allows us to ask whether a particular, initially rare, strategy can successfully invade a metapopulation of another strategy (Methods).” (previously: “We embed these competitions within a broader framework of evolutionary game theory[55,64,65]“)

4) Are the initial frequencies of the two competing populations important for the final outcome? This is related to the previous question regarding different outcomes between local and global. Also, while the model is currently deterministic, in real case scenarios noise plays a big role, so it would be good to briefly discuss this.

This question is covered by one of several parameter sweeps that were in the original submission (Figure S6). There we study the effects of initial frequency in a model of stochastic variation in the initial strain frequencies, using the most general model where all strategies compete with one another. Our results show our predictions are robust for different initial frequencies in the two strains. We have now extended our description of this result in the main text (section "The coevolution of regulated attack strategies') to make this clearer.

5) In the regulated models, the parameter constraints allow f_induced and f_initial to range between 0 and 1. But because of how the model is set-up, this means that it can happen (and it definitely does looking at Figure 2) that f_initial+f_induced=f is outside the [0,1] range, which is though the constrained range for the f of the constitutive competing strain. This might cause strange model behavior and definitely an unfair competition, so these parameters should be removed/checked.

We are sorry about this confusion in the text. To be clear about the definitions: f_initial indicates the toxin investment at the initial condition, f_induced indicates the level of toxin investment when a sensor is triggered. Thus, the change in investment occuring due to the trigger is f_induced-f_initial. We clarify this now in the text and in the legend of Figure 2.

1) The model developed in the manuscript captures the toxin regulatory system in bacteria, which is very interesting. However, the title of the manuscript and the abstract should be revised to better reflect the specific system under study.

This is a good suggestion, and we have now revised the abstract and title accordingly to be more specific about the types of regulation (quorum sensing and stress responses) and the phenotype under regulation (toxin production).

2) The population dynamic equations in eq set 1 could be analysed further to get some analytical handle to promote our understanding. The authors could start from the following paper Vasconcelos, P., Rueffler C., 2020, How Does Joint Evolution of Consumer Traits Affect Resource Specialization? The American Naturalist 195: 331-348.In addition, the final biomass densities used in the calculation of the invasion fitness (Eq. 2) could be further simplified by focusing on the equilibrium solutions of Eqs. 1 (perhaps under which assumptions the solutions are possible could already be interesting). Providing an expression for the invasion fitness would be a real plus.

Thank you for this suggestion, we looked through this paper and considered it as a starting point but it does not capture our problem well and, as discussed above, analytics more generally are incapable of capturing key features of our system.

3) Please clarify the rationale for choosing parameters in the simulations, and discuss robustness, beyond the statement "parameters of the algorithm […] are chosen to achieve short simulation times and good convergence behaviour as determined by visually inspecting the distribution of population parameters over time." (727-730)

As is typical in non-adaptive genetic algorithms, we base our choice of the control parameters of the algorithm on positive convergence behaviour and short convergence times (Melanie 1996). Further, we reran the simulations with 20 different sets of hyper-parameters to confirm that our findings are not specific to the set of parameters we used initially. We now add this explanation into the "Genetic Algorithm" section: "In our sensitivity analysis, we also examine the results of the genetic algorithm with alternative sets of control parameters, including a smaller and a larger size of the mutation standard deviation (sd=0.01 and sd=0.0005), a smaller and larger proportion of “migrating” strategies in each generation (5 of 60, and 20 of 60), and five different sizes of the population of strategies (50, 70, 80, 90, 100). This yields 20 alternative parameters combinations." (see "Genetic Algorithm" section). The results were consistent with our main simulation. We have also extended the appropriate sentence in the Results section: "Despite a great number of potential combinations (over two million different competitions), and with different sets of hyperparameters of the genetic algorithm, we again see a clear winner in toxin-based regulation, both for our normal parameters (Table 1, Figure 4c) and for sweeps that consider broad ranges of these parameters (Figures S5) and a wide range of initial frequencies of the two strains (Figure S6). "

4) Some of the findings are intuitive, as the authors acknowledge. E.g. regulated traits outperform non-regulated traits, but others are more surprising and very interesting. For example, many toxins are quorum-sensing regulated (e.g. phenazines in *P. aeruginosa*). But the authors show that this is not an ideal mechanism because a strain might be killed before it's reaching a high enough density to switch on QS-regulated toxins. I'd like to see a more detailed discussion on this putative mismatch. Related to this, my intuition is that the three mechanisms might work in concert, i.e. a toxin is QS controlled, but the threshold for QS activation is lowered by competition sensing. This would be interesting to discuss.

We agree, these are two very interesting points. We have now included a more detailed discussion of why QS might be more common than we are expecting from our results. We now say (in "Conclusion" section) "However, if detecting damage is the best basis for attack, why do some bacteria use these other forms of regulation? For short competition times, our model predicts that the three regulatory strategies are largely equivalent (Figures 3 and S7). A short duration of competition between strains removes the benefit of decreasing toxin production once an attacker has been overcome. Under these conditions, the evolutionary path to one form of regulation may largely be determined by differences in costs for regulatory networks and which pre-existing regulatory systems are available for co-option [98,99]. We predict, therefore, that mechanisms to reciprocate attacks are particularly valuable in environments where warfare commonly leaves a victor unchallenged for a long time afterwards. Consistent with this, one of the clearest examples of reciprocation occurs in *E. coli* [66,84,88], which uses colicin toxins to displace other strains and persists for long periods within the mammalian microbiome [100].

Another possible explanation for why some bacteria do not use cell damage to regulate their toxins comes from the notion of ‘silent’ toxins. These are toxins that are not easily detected by the cell’s stress responses, which may limit the potential for a toxin-mediated response. For example, some toxins depolarise membranes[101] and may be favoured by natural selection specifically because they do not provoke dangerous reciprocation in competitors [87].".

We have also extended our discussion on how multiple sensory mechanisms might work in concert ("Conclusion" section): "[…] bacteria appear to use multiple forms of regulation in order to integrate information from multiple sources [41]. For example, *Streptomyces coelicolor* regulates antibiotic production via both nutrient limitation [102] and mechanisms that detect incoming antibiotics (envelope stress [103]). A potential future use of our modelling framework would be to study how these combined regulatory strategies evolve."

5) The conclusion on lines 32-35 is strong. It might apply to mixed conditions as simulated here. However, in spatially structured habitats reciprocal fighting might lead to co-existence as competitors manage to defend their 'territories' and fighting only occurs at the borders. The swift elimination of competitors is maybe less common than assumed in real-world set ups.

We agree with this, and have now removed this sentence from our abstract.

6) The argument is that toxin sensing works best as the winner switches off toxin production once the competitor is eliminated. However, toxins might outlast their producers, such that the switching off might take longer than assumed. Can the authors comments on this?

Yes, in our local competitions toxins can outlast their producers. Consistent with this, winning strains commonly downregulate their toxin once the competitor toxin falls to a value that is small, but not exactly zero. That is, these strategies use low rather than zero toxin as a cue for elimination of the opponent. In response, we have altered our discussion of this as follows: "Instead, the winning strategies are initially aggressive and only become passive once an opponent is eliminated (Figure 4 d-f) – or nearly eliminated, because eliminated strains can be outlasted by their toxins, which in turn will take a while to be lost entirely. Indeed the winning strains in these competitions downregulate their toxin once the competitor toxin falls below a value that is small but still non-zero."

[Editors’ note: what follows is the authors’ response to the second round of review.]

Essential revision:While the motivation for the local vs. global analysis is well explained, the results from this analysis should be clarified. In figure 1, interesting examples are proposed of strategies that might win/lose locally and then lose/win globally. These are very interesting cases that point at non-trivial competition dynamics. It would be important to add a figure or table (perhaps in the supplement) and an associated discussion paragraph where it is shown that doing this additional meta-population analysis is necessary and adds something to the local competition part. The aim of this addition would be to address the following question: Would the optimal strategy change if one didn't do the global competition step? Some statistics of counterintuituive scenarios explored, based on their classification in figure 1d, would address this point.Reviewer #1 (Recommendations for the authors):The manuscript puts forward exciting hypotheses about the strategies of toxin production in bacteria, which would be very interesting to test experimentally. Also, future work that tries to shed deeper analytical insight in the results would undoubtedly be very interesting.I think the work is definitely worth publication. I find the presentation of the results sometimes difficult to follow, partially because the details of the models, which is very helpful to understand the results, is detailed at the end of the paper in the methods. But probably this is unavoidable given the format.I believe the current version of the manuscript reads better and clarifies some of the misunderstandings in previous versions. There are, however, some criticisms that have not, in my opinion, been well addressed.1) I appreciate the complexity of the model and the strengths that come with including this complexity. The new analytical work carried out to investigate stability of the fixed points helps towards the analytical interpretation of the results. I think, however, the criticism previously raised by the reviewers was trying to determine whether a simpler model, analytically tractable, would be able to reproduce some of the results showed here, while giving more analytical insight. I don't think the manuscript currently addresses this issue. On the other hand, I also think that it can be left to future work as it is a very interesting and challenging research direction.2) I am satisfied with the motivation behind the local versus global analysis, which I think is very important, especially in potential future applications to spatial settings of this work. The motivation behind the analysis was never in question. What I don't understand are the results from this analysis. In figure 1, interesting examples are proposed of strategies that might win/lose locally and then lose/win globally. These are very interesting cases that point at non-trivial competition dynamics that would be interesting to investigate further in future work, but I don't see these cases discussed anywhere in the results. I understand that the results of the algorithm come from a sequence of local competition, dispersal, seeding, etc…, that include this metapopulation competition, but I would like to see a figure/paragraph/discussion where it is shown that doing this additional meta-population analysis adds something to the local competition part. Would the optimal strategy change if one didn't do the global competition step? If the answer is yes, which I imagine it is, where do I see this?

We were very pleased that the referees were enthusiastic about publication. As requested, we have now addressed the remaining concerns about the non-trivial competition dynamics shown in Figure 1d. As suggested, we have added a Supplementary Table that reports on the occurrence of all four possible competition dynamics (as per Figure 1 d) occurring in the simulations underlying Figure 2. This shows that all of the four possible outcomes occur. The less intuitive outcomes are rarer than the more intuitive ones, as one would expect, but the fact that all cases occur underlines that the correct metric for fitness is one that accounts for global competition as well as local competition. We now highlight this analysis in the description of the Figure 1, and we also discuss the new Table where we describe our findings of Figure 2.